# Macrophage-associated wound healing contributes to African green monkey SIV pathogenesis control

Fredrik Barrenas[1,2,16], Kevin Raehtz[3,4,16], Cuiling Xu[5], Lynn Law[6,7], Richard R. Green[6,7], Guido Silvestri[8,9], Steven E. Bosinger[8,9], Andrew Nishida[1], Qingsheng Li [10], Wuxun Lu[10], Jianshui Zhang[10], Matthew J. Thomas[6,11], Jean Chang[6,7], Elise Smith[6,7], Jeffrey M. Weiss[1], Reem A. Dawoud[8], George H. Richter[5], Anita Trichel[12], Dongzhu Ma[13], Xinxia Peng [14], Jan Komorowski [2,15], Cristian Apetrei[3,4], Ivona Pandrea[4,5] & Michael Gale Jr [6,7,11]*

Natural hosts of simian immunodeficiency virus (SIV) avoid AIDS despite lifelong infection. Here, we examined how this outcome is achieved by comparing a natural SIV host, African green monkey (AGM) to an AIDS susceptible species, rhesus macaque (RM). To asses gene expression profiles from acutely SIV infected AGMs and RMs, we developed a systems biology approach termed Conserved Gene Signature Analysis (CGSA), which compared RNA sequencing data from rectal AGM and RM tissues to various other species. We found that AGMs rapidly activate, and then maintain, evolutionarily conserved regenerative wound healing mechanisms in mucosal tissue. The wound healing protein fibronectin shows distinct tissue distribution and abundance kinetics in AGMs. Furthermore, AGM monocytes exhibit an embryonic development and repair/regeneration signature featuring TGF-β and concomitant reduced expression of inflammatory genes compared to RMs. This regenerative wound healing process likely preserves mucosal integrity and prevents inflammatory insults that underlie immune exhaustion in RMs.

[1] Department of Microbiology, University of Washington, Seattle, WA, USA. [2] Department of Cell and Molecular Biology, Uppsala University, Uppsala, Sweden. [3] Division of Infectious Diseases, Department of Medicine, University of Pittsburgh, Pittsburgh, PA, USA. [4] Department of Microbiology and Molecular Genetics, School of Medicine, University of Pittsburgh, Pittsburgh, PA, USA. [5] Department of Pathology, School of Medicine, University of Pittsburgh, Pittsburgh, PA, USA. [6] Department of Immunology, University of Washington, Seattle, WA, USA. [7] Center for Innate Immunity and Immune Diseases, University of Washington, Seattle, WA, USA. [8] Department of Pathology & Laboratory Medicine, Emory University, Atlanta, GA, USA. [9] Division of Microbiology & Immunology, Yerkes National Primate Research Center, Emory University, Atlanta, GA, USA. [10] Nebraska Center for Virology, School of Biological Sciences, University of Nebraska-Lincoln, Lincoln, NE, USA. [11] Washington National Primate Research Center, University of Washington, Seattle, WA, USA. [12] Divison of Laboratory Animal Resources, University of Pittsburgh, Pittsburgh, PA, USA. [13] Department of Orthopedic Surgery, University of Pittsburgh, Pittsburgh, PA, USA. [14] Department of Molecular Biomedical Sciences, North Carolina State University, Raleigh, NC, USA. [15] Institute of Computer Science, PAN, Warsaw, Poland. [16] These authors contributed equally: Fredrik Barrenas, Kevin Raehtz. *email: mgale@uw.edu

Curbing the HIV pandemic remains a global health priority, but significant scientific hurdles prevent an effective vaccine or therapy[1–3]. To this end, defining the molecular HIV/SIV immunopathogenic mechanisms remains critical. The most useful model systems to this end are possibly the African non-human primate (NHP) species that carry SIV in the wild, termed natural SIV reservoir hosts, e.g. the African green monkey (AGM—*Chlorocebus* genus), sooty mangabey (SM—*Cercocebus atys*) and mandrill (*Mandrillus sphinx*). Remarkably, these species remain disease-free after SIV infection despite high levels of viremia and major mucosal CD4+ T cell depletion[4–6].

In contrast, SIV-infected rhesus macaques (RMs) and other Asian NHPs exhibit sequelae typical of AIDS including CD4+ T cell loss, lymphoid tissue fibrosis, hypercoagulation, intestinal epithelium breakdown, and opportunistic infections[7,8]. Comparative studies of natural and pathogenic SIV hosts have delineated several immunological and genomic features as novel treatment targets, aimed at reducing harmful inflammation during HIV infection[5,9].

The defining trait of the natural SIV host phenotype is lack of protracted immune activation[10–13]. In infected humans and macaques, markers of immune activation are strong indicators of mortality[14]. Immunological mechanisms linked to the avoidance of pathogenic immune activation in natural hosts include: (1) alterations in sequence or expression of CD4[15] or CCR5[16,17] on target cells, restricting infection in vulnerable lymphocyte populations; (2) resolution of interferon signaling and innate immune activation;[10,18,19] (3) maintenance of mucosal integrity[7,15,18,20,21]. Additionally, (4) macrophages, which are prevalent at SIV/HIV infection sites[22], are less susceptible to SIV in natural hosts[5,23].

Crucial pathogenic events occur within days of SIV infection, including seeding of viral reservoirs[3] and systemic immune activation[24]. In RMs, the intestinal tract undergoes dramatic loss of immune cells and tight junction protein expression during the acute infection[25], partly due to macrophage SIV infection[26]. This allows microbial components to penetrate the epithelial barrier, aggravating inflammatory responses[7]. Transcriptomic analyses of RMs acutely infected with SIV found activation of genes associated with NK cells, monocytes and T cells in rectal tissues[24], underscoring the prevalent hypothesis that AIDS starts with a massive insult to the mucosal immune structures, leading to persistent inflammation and systemic immune defects[20]. This hypothesis describes the first days after SIV/HIV infection as critical for the immune programming driving infection outcome.

In natural SIV hosts, transcriptional studies of acute SIV infection have utilized intravenous inoculation and focused on peripheral blood immune cells. In SIV-infected AGMs, CD4+ cells mount interferon responses earlier than in RMs, but resolve them after the acute phase, exemplifying how early innate immune programming could potentially drive nonpathogenic infection outcome[18,19,27]. Similarly, both colon and lymph nodes show strong but transient immune activation, Th1 polarization and progenitor cell regeneration during acute infection in AGMs[27]. Intestinal damage in SIV-infected AGMs, recapitulates key features of pathogenic SIV infection such as microbial translocation, increased viral load, and systemic immune activation[28,29]. Myeloid DCs also show diminished activation and apoptosis during acute SIV infection in AGMs, compared to pathogenic hosts[30]. This emphasizes the importance of mucosal immune actions during acute SIV/HIV infection to achieve viral and disease control in natural hosts.

To define the gene expression signatures of acute SIV/HIV infection that link with mucosal immune programming and systemic immune regulation, we conduct parallel examinations of AGM and RM transcriptional signatures and tissue responses after the intrarectal virus challenge. Over a time-course covering acute SIV infection, total RNA-seq, bioinformatic, virologic, and histologic analyses are performed. We report highly dissimilar expression dynamics between the two SIV hosts. To fully interrogate these differences, we develop a systems biology approach, referred to as Conserved Gene Signature Analysis (CGSA), based on correlating expression signatures in acute SIV infection across informative transcriptomic datasets from the public domain, including across diverse species. These datasets entail relevant tissues (e.g. mucosal tissue, epithelial cells) and conditions (e.g. tissue damage, cytokine stimulation), such that CGSA enables data-driven, unbiased discovery of pathogenic or protective mechanisms of SIV disease progression.

Using CGSA, we identify a strong wound healing signature in the AGM model, including monocyte regulated tissue remodeling similar to that of the axolotl, a neotenic salamander species with remarkable wound healing abilities. In contrast, RMs induce microbial pattern recognition receptor and inflammatory signaling. Our observations implicate monocyte-directed mucosal immune programming in AGM SIV responses, that concomitantly maintain intestinal integrity and avoid microbial activation.

Thus, maintenance of mucosal integrity through control of monocyte immune responses, could be a valuable therapeutic strategy to avoid systemic immune activation that underlies SIV/HIV disease progression and AIDS.

## Results

**SIV inoculation and rectal tissue transcriptomic profiling**. To identify the host responses that determine pathogenic or natural infection outcomes, 28 AGMs and 24 RMs underwent a single high-dose intrarectal inoculation with two genetically diverse swarm stocks ($SIV_{sab}92081$ and $SIV_{mac}251$) previously shown to have comparable infection kinetics[11,31]. As several SIVs replicate poorly in non-adapted hosts[10,32], we selected viruses to ensure comparable viral loads, and natural or pathogenic infection outcomes in AGMs and RMs, respectively.

Tissue samples were collected upon necropsy at specific time points over an acute infection course: pre-viremic (D1–4), early viremic (D5–6), viral peak (D9–12) and viral set point (D50 or D84) (Fig. 1a). Pre-inoculation baseline samples were collected by pinch biopsy. In collected blood samples, plasma was separated to measure viral load. In both infection models, peak viral load occurred at D9–12 ($1.39 \times 10^7$ in AGMs, $2.48 \times 10^7$ in RMs, Fig. 1b). At set point, viral load in AGMs was approximately one log lower than RMs ($6.95 \times 10^4$ and $5.69 \times 10^5$, respectively, $p = 0.023$ by two-sided $T$-test). Viral loads in AGM and RM rectal tissues were comparable (Supplementary Fig. 1).

**Contrasting responses to acute SIV infection in AGMs and RMs**. Rectal tissues collected during necropsy and pre-inoculation underwent total RNA-seq profiling. The resulting sequence reads were mapped to the genomes of the respective NHP species for differential expression (DE) and other bioinformatic analyses. Direct comparisons between the two species utilized genes with human ENSEMBL one-to-one orthologs in both RMs and AGMs. DE analysis was performed in two ways: firstly, for each time point, pre-challenge pinch biopsies were used as baseline comparators for the rectal necropsy samples, resulting in hundreds of DE genes at each time point with fewer than 10% being shared between the species at most time points (Table 1). Secondly, to avoid potential bias due to different sample types as baseline, DE analyses were performed using pairwise comparisons between all time points (all-to-all comparisons) for each animal model which resulted in a union of 7420 DE genes between the two species (Table 1). In summary, the DE analyses indicated a

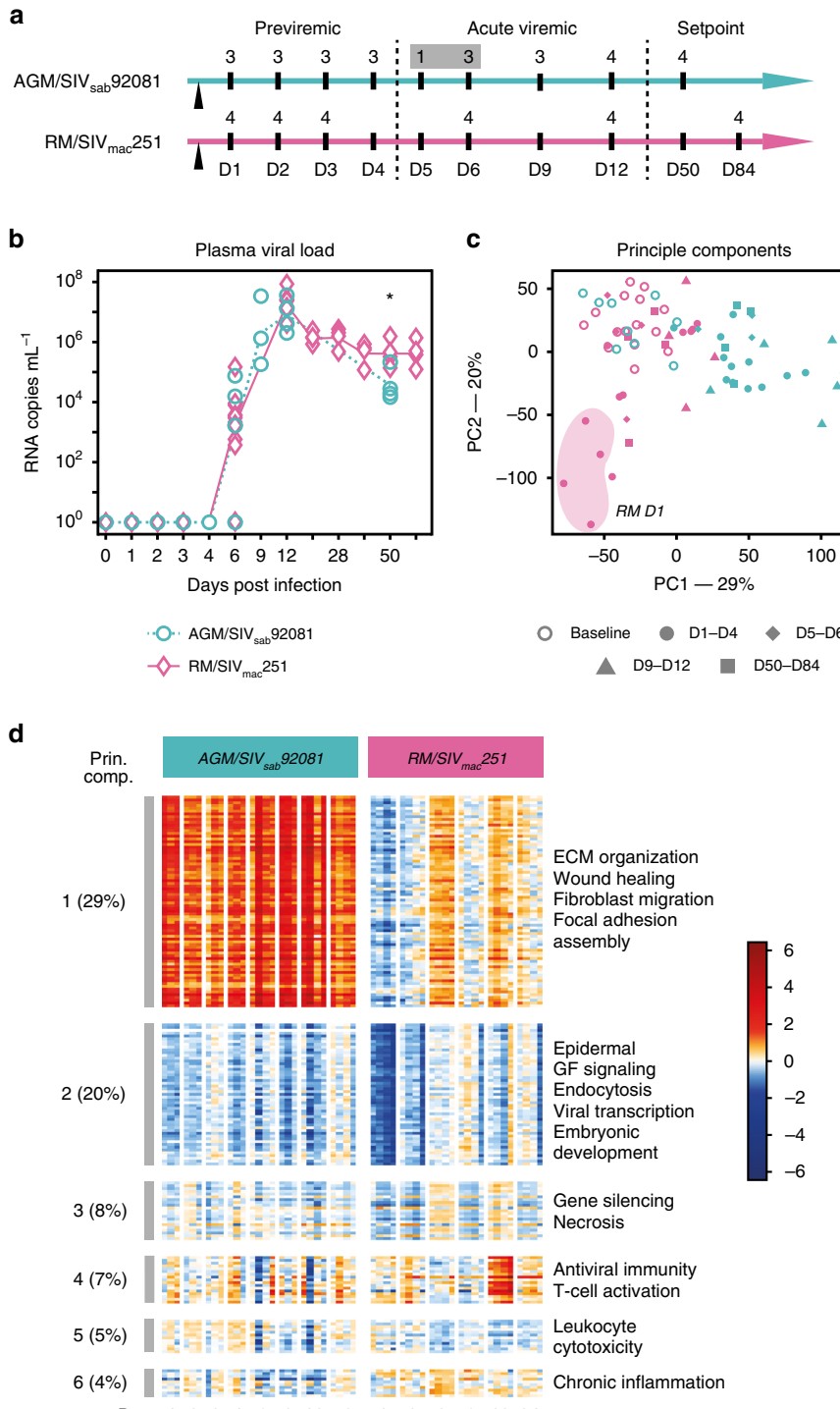

**Fig. 1** Study overview. **a** Study design. Numbers indicate the number of necropsied animals at each time point. **b** Plasma viral load in AGMs and RMs, given as log$_{10}$ values. Significant difference ($p < 0.05$, two-sided $T$-test) is indicated by a star. **c** Principle component (PC) analysis of the 7420 DE genes, showing the strongest contrasts between AGM and RM time dynamics. Numbers at the axis indicate the percentage of total variation explained. **d** Heatmap showing representative gene sets obtained by PC analysis. From the top six principal components, we show the genes with the strongest contribution to variation in proportional numbers. A functional enrichment test was performed on these gene sets; the top enriched functions are listed on the right of the heatmap

sharp contrast in the host response to infection between the two species.

A principle component (PC) analysis (Fig. 1c, d) showed that the AGMs mounted an early, persisting response to SIV infection (first PC). The D1–D4 response in the RM model, by contrast, was transient, subsequently returning towards baseline (second

PC). Gene ontology (GO) enrichment revealed that the first PC (AGMs) showed upregulation of extracellular matrix (ECM) organization, wound healing, fibroblast migration and focal adhesion assembly. The second PC (RMs) showed transient downregulation of epidermal growth factor (GF) signaling, endocytosis, viral transcription, and embryonic development.

**Table 1 Study overview[a]**

| Time point | AGM N | AGM Up | AGM Down | RM N | RM Up | RM Down | Shared[b] Up | Shared[b] Down |
|---|---|---|---|---|---|---|---|---|
| D1 | 3 | 1703 | 687 | 4 | 903 | 2058 | 117 | 270 |
| D2 | 3 | 690 | 71 | 4 | 231 | 893 | 41 | 17 |
| D3 | 3 | 535 | 40 | 4 | 651 | 241 | 197 | 1 |
| D4 | 3 | 1035 | 422 | 0[c] | – | – | – | – |
| D5 | 1 | 2241 | 525 | 0[c] | – | – | – | – |
| D6 | 3 | Na[d] | Na[d] | 4 | 28 | 195 | 25 | 9 |
| D9 | 3 | 2265 | 1063 | 0 | Na[c] | Na[c] | Na[c] | Na[c] |
| D12 | 4 | 1707 | 526 | 4 | 729 | 575 | 299 | 77 |
| D50/D84 | 4 | 905 | 82 | 4 | 180 | 267 | 93 | 7 |

[a]Differential gene expression analysis was perform as described in the Methods. Number of animals, number of up- and down-regulated genes for each model is given
[b]The number of genes that were shared between the two animal models at each time point
[c]No RMs were necropsied at these time points
[d]D5 and D6 were merged in the AGM study

surprisingly, the PCs enriched in typical immune responses (fourth, fifth and sixth PC) comprised only 16% of the dataset's total variability. These gene sets were associated with Antiviral Immunity, T and B cell activation and chronic inflammation. The fourth PC was strongly induced at D12 in RMs, corresponding to the viral peak. (Fig. 1d). A comprehensive catalog of DE genes, and functional enrichment results can be found in Supplementary Data 1.

**Systems biology approach to characterize acute SIV responses.** As the initial analysis identified thousands of genes with divergent time kinetics between the two SIV models, we reasoned that the infection outcome is determined by specific cellular responses, occurring on specific days. To understand these kinetics, we devised CGSA as a systems biology approach based on correlating acute SIV infection data with informative time series datasets from the public domain (Fig. 2a). This data-driven approach adds time dimensions to documented, static gene functions, avoids false positive gene–gene interactions and adds statistical robustness.

For CGSA we collected transcriptomic datasets (Reference Datasets) pertaining to GO Biological Processes enriched among genes identified during PC analysis, resulting in a catalog describing four biological processes: wound healing (skin or mucosal tissues sampled post wounding), experimental colitis (colon tissues sampled post oral dextran sulfate sodium or dinitrobenzene sulfonic acid administration), in vitro cytokine stimulation of epithelial cells, and microbial colonization (colon tissues collected from germ-free mice after introduction of normal intestinal microbiota). We also included datasets from large-scale clinical studies of colon cancer and inflammatory bowel diseases. These six biological processes entailed 23 reference datasets, amounting to 2438 microarray samples (Table 2, Supplementary Fig. 2).

We then constructed a co-expression network (the Acute SIV Co-expression Network) using our acute SIV infection RNA-seq data, thereby identifying gene pairs that were co-expressed in both species. Each gene–gene interaction in the network was assigned a weight, based on its correlation coefficient in the acute SIV infection data and in the 23 reference datasets. Finally, a community detection algorithm partitioned the 7420 genes in the network, into functional gene modules related to specific biological processes, resulting in 23 network modules, referred to as modules A-W (Fig. 2b, c). (The number of network modules and reference datasets was the same by coincidence.) The differential gene expression across these functional modules showed remarkable contrasts between the AGMs and RMs. For

instance, genes in modules A through E were frequently upregulated in the AGM model only. Biological processes enriched in these modules included cell polarity, cell–ECM adhesion, gap junction assembly, TOR signaling, and Nuclear transport. The genes in modules I-N were generally down-regulated in RMs and virtually unchanged in AGMs. These modules were enriched for lipid metabolism, desmosome organization, microtubule severing, TLR signaling, DNA replication, and epithelial cell–cell adhesion. The Acute SIV Co-Expression Network constitutes the most comprehensive comparison of host responses to acute SIV infection between a natural and a pathogenic host model to date. For complete results, see the Supplementary Data.

To determine if transcriptomic signatures from the reference datasets could produce the stark differences between the two SIV models, we examined CGSA on a genomic scale. Specifically, we calculated the Pearson correlation coefficient between the gene expression log_2 fold change (logFC) values from acute SIV infection at each time point, to each reference dataset (Fig. 2d).

This revealed that at the viral replication peak, the transcriptional profile in RMs was correlated with microbial colonization, cytokine stimulation and, importantly, tissue damage. These conditions included: the inflammatory stage of wound healing (Fig. 2d, Wound Healing 24 h), a period of strong neutrophil and M1 macrophage activation[33], and chemically induced colitis (Fig. 2d, Colitis Models). In contrast, the gene expression profile in AGMs moderately correlated to colitis models and cytokine stimulation (including type I and II interferons, but not TNF-α) from the previremic stage, throughout the course of infection. However, correlations to wound healing 24 h and microbial colonization were negligible or negative, indicating that AGMs mount certain immune pathways during the previremic stage, but even during peak viremia, immune activation and tissue damage are limited. AGMs also showed stronger correlation to late stage wound healing (day 7) in the reference datasets. By day 7, the inflammatory wound healing stage is normally resolved and tissues are undergoing re-epithelization, angiogenesis, and remodeling[34].

**Absence of microbial signature in AGM innate immune response.** The genome-wide comparison between acute SIV and reference datasets showed a stronger association between RM expression profiles and inflammatory tissue damage, at peak viremia. To reveal the genes that drive these contrasts, we systematically removed each of the 23 gene modules from the Acute SIV Co-Expression Network and re-calculated the correlation to the reference datasets in the AGMs and RMs separately. A

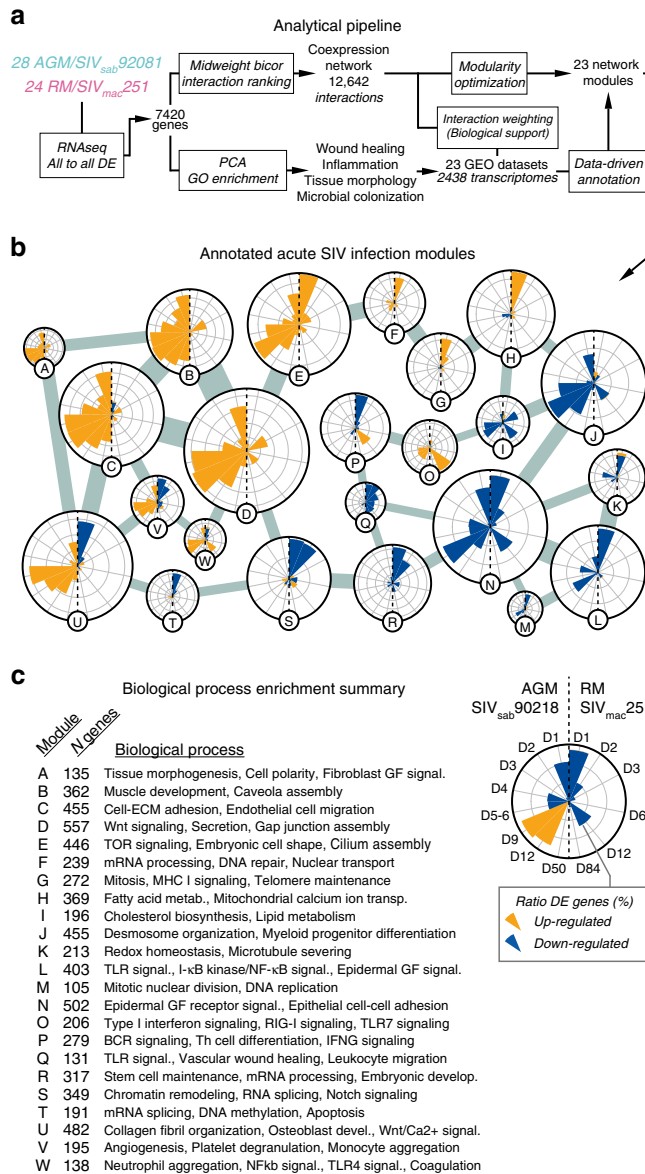

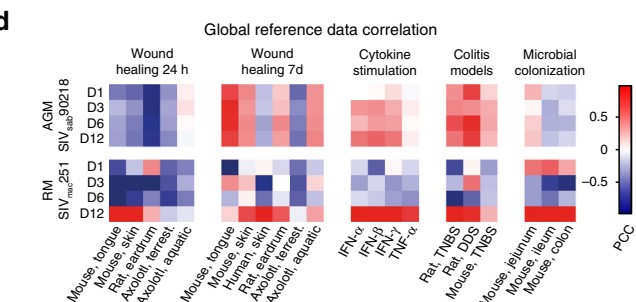

**Fig. 2** CGSA bioinformatic pipeline overview. **a** Systems biology approach to improve identification and annotation of the Acute SIV Co-Expression Network in acute SIV infection data. (Lower path) GO enrichment analysis of the 7420 DE genes was used to identify biological function relevant for acute SIV infection in AGMs and RMs. Twenty-three reference datasets pertaining to relevant processes were collected and used to improve the biological support and annotate the Acute SIV Co-Expression Network. (Upper path) A co-expression network was first constructed from acute SIV infection data. Each interaction was then weighted based on correlations in reference datasets. The network was then partitioned into modules that underwent annotation based on reference datasets. **b** The Acute SIV Co-Expression Network, in which the resulting modules were connected to the two modules with which it shared the highest number of interactions. Each pie chart shows the ratio of up (orange) and down (blue) regulated genes at each time point in each SIV host species. Concentric circles mark 20% of the total number of genes in each module. Node sizes represent numbers of genes in each module. Interaction thickness represent the number of interactions between each module. **c** Summary of functional enrichment tests for each network module. **d** Whole-genome correlation between acute SIV infection and reference datasets. Colors in heatmaps represent the Pearson correlation coefficient between logFC values in acute SIV infection and the reference dataset specified. In wound healing, two points from the time series, 24 h and day 7, are shown. In cytokine stimulation, 12 h (interferons) or 6 h (*TNF-α*) data are shown. In colitis and microbial colonization, day 7 and day 16, respectively, are shown

RMs, and inflammatory reference datasets (24 h wound healing, cytokine stimulation, microbial colonization). It also had a moderate impact on the correlation to experimental colitis (Fig. 3a). The corresponding effect in AGMs was low or negligent. The Interferon Module was primarily upregulated around the viral peak (from D6 to D12, of all DE genes from the Interferon Module, 93% were upregulated in AGMs and 98% in RMs). Compared to RMs, AGMs activated the interferon module earlier, with 78 DE genes at D6, and highest fold change in the antiviral immune genes MX1, IFIT1, C4BPA, and IFI6. At D12, fewer genes were DE in AGMs (87) than RMs (171). Only four genes were unique to AGMs (RNF25, NKX3-1, GPR128 and DPYS). The module removal analysis suggested that in AGMs, the interferon module was specifically activated by innate immune signaling (Fig. 3a, cytokine stimulation), while in RMs the module was also affected by bacterial PAMPs and tissue damage (Fig. 3a, wound healing 24 h and microbial colonization). To confirm this independently, we used Ingenuity pathway analysis (IPA), based on manual curation of scientific publications, to identify upstream regulators of DE genes from AGMs and RMs, separately. We performed Fisher's exact tests between targets of each IPA upstream regulator, and DE genes in AGMs and RMs, using the interferon module as background (Fig. 3b). This analysis supported a regulator role for the bacterial component LPS in RMs, but not AGMs, whereas interferon-γ (*IFN-γ*) and interferon-α (*IFN-α*) type interferons had similar *p*-values for the two model systems. The AGM response showed stronger enrichment for interferon-λ (*IFN-λ*), a potent antiviral cytokine[35], and growth factor prolactin (*PRL*) which shows elevated levels in patients with HIV[36]. We repeated the enrichment analysis on biological functions, confirming stronger activation of antiviral and antimicrobial immune responses in RMs, while only AGMs showed activation of leukocyte and myeloid cell recruitment (Fig. 3c). This outcome agrees with the upstream regulator analysis, as several myeloid cell types, including monocyte-derived macrophages and dendritic cells, produce high levels of *IFN-λ* in response to viral infection[37]. *IFN-λ* inhibits HIV infection of macrophages, potentially contributing to the low macrophage SIV infection in natural host species[38].

reduction in the Pearson correlation coefficient would indicate a link between the module and reference dataset. Removing single modules did not substantially reduce correlation between acute SIV infection and reference data in most cases, showing that these correlations were driven by large gene sets with diverse functions. The notable exception was module O, a 206 gene module associated with Type I interferon signaling, RIG-I signaling, and TLR7 signaling.

Removing module O (referred to as the interferon module) had a strong impact on correlations between D12 SIV infection in

**Table 2 Reference datasets**

| GSE | Description | Species | Tissue | $N_{samples}$ | PMID |
|---|---|---|---|---|---|
| GSE23006 | Wound healing | Mouse | Skin | 24 | 20704739 |
| GSE23006 | Wound healing | Mouse | Tongue | 24 | 20704739 |
| GSE28914 | Wound healing | Human | Skin | 25 | 23082929 |
| GSE17698 | Wound healing | Rat | Eardrum | 40 | 21919009 |
| GSE35255 | Wound healing | Axolotl, aquatic | Skin | 16 | 22485136 |
| GSE35255 | Wound healing | Axolotl, terrestrial | Skin | 16 | 22485136 |
| GSE9293 | Colitis by TNBS | Rat | Colon | 18 | – |
| GSE9281 | Colitis by DSS | Rat | Colon | 42 | – |
| GSE35609 | Colitis by TNBS | Mouse | Colon | 34 | 23226271 |
| GSE19392 | IFN-β stimulation | Human | Airway epithelial cells | 169 | 20064372 |
| GSE19182 | Cytokine stimulation | Human | HNEC cells | 21 | 22005912 |
| GSE27870 | TNF-α a stimulation | Human | Endothelial cells | 24 | 22121215 |
| GSE32513 | Microbial colonization | Mouse | Jejunum | 48 | 22617837 |
| GSE32513 | Microbial colonization | Mouse | Ileum | 48 | 22617837 |
| GSE32513 | Microbial colonization | Mouse | Colon | 48 | 22617837 |
| GSE51269 | TLR knock-out | Mouse | Colon | 20 | 25210121 |
| GSE39582 | Colon cancer | Human | Colon | 585 | 23700391 |
| GSE26253 | Colon cancer | Human | Colon | 432 | 24598828 |
| GSE44076 | Colon cancer | Human | Colon | 246 | 25215506 |
| GSE17536 | Colon cancer | Human | Colon | 177 | 19914252 |
| GSE11223 | Ulcerative colitis | Human | Colon | 202 | 18523026 |
| GSE20881 | Crohn's disease | Human | Colon | 172 | 20848455 |
| GSE48634 | Ulcerative colitis | Human | Colon | 171 | 25171508 |

Before the viremic stage, we found little genome-wide correlation between inflammatory processes in reference datasets (acute wounding, cytokine stimulation, colitis, microbial colonization) and SIV infection. However, several events that contribute to pathogenesis are pre-viremic, e.g. reduction of tight junction protein expression at day 3[25]. We, therefore, performed an upstream regulator analysis on the Interferon Module to examine if the LPS signature was present at day 3 in RMs (Fig. 3d). While the LPS signature was indeed present ($p = 0.0038$), there was no enrichment of *IFN-α*, *IFN-γ* or *IFN-λ* targets. AGMs, by contrast, showed a stronger enrichment for targets of *IFN-α*, *IFN-γ*, *IFN-λ*, and *PRL*, supporting an early antiviral response in the absence of microbial translocation. The functional categories at day 3 (Fig. 3e) showed that AGMs had already activated myeloid cell migration. In addition, even though the Interferon Module was specifically regulated by interferons, the AGM DE genes within this module were associated with tissue development (adipose, epithelial and muscle tissue).

**Early signatures of non-inflammatory wound healing in AGM.** The genome-wide analysis showed an inflammatory tissue damage signature in RMs, while AGMs activated a signature similar to later stages of wound healing. These observations indicate that unlike RMs, AGMs can limit tissue damage during acute infection by rapidly repairing SIV-induced wounds before they cause colitis-like damage and microbial translocation. This process would help control protracted inflammation and maintain mucosal T cell homeostasis[28]. To investigate the gene signatures linked to these outcomes, we explored wound healing mechanisms in the Acute SIV Co-Expression Network and reference datasets.

The six wound healing datasets entailed biopsies from healing skin or mucosal tissue in human, mouse or rat, as well as axolotl (*Ambystoma mexicanum*). Aquatic axolotl wound healing is non-inflammatory and results in fully functional, scar-free tissue[39]. A terrestrial species was used as comparison, as they exhibit a more mammal-like repair process[40]. Using this data, we created a unique resource of wound healing signatures from different species, time points and tissues in acute SIV infection. First, we

defined interactions in the Acute SIV Co-Expression Network with a high correlation to wound healing reference datasets using a permutation test; the correlation coefficients from each wound healing dataset were randomly reordered among the network interactions 100,000 times. A sub-network was then defined from interactions with an average correlation score from the wound healing datasets above 95% of random permutations. Second, we removed genes that were DE in less than two wound healing datasets. The resulting network, referred to as the Wound Healing Network, contained 191 interactions between 166 genes (Fig. 4a). The Wound Healing Network consisted of two large clusters, interconnected by two interactions, one between LAMB1 and LAMC1 and one between SPARC and THY1, implying that these interactions are likely critical for the wound healing process.

CGSA further revealed that the genes in the Wound Healing Network showed consistent expression dynamics across all reference datasets, except the aquatic axolotl. These dynamics entailed downregulation during the first 24–72 h, corresponding approximately to the inflammatory stage of normal mammalian wound healing, followed by upregulation of genes during the later tissue remodeling and re-epithelization stage[41] (Fig. 4b). In the aquatic axolotl, the Wound Healing Network genes stayed near baseline expression level during the mammalian inflammatory stage, but showed strong upregulation during remodeling at day 7. In RMs, the Wound Healing Network genes were downregulated during the first days post-infection, and then returned to baseline. This pattern suggests that the RM host response to acute SIV infection involves the inflammatory stage of wound healing, but not the tissue remodeling (Fig. 4c). In stark contrast, AGMs showed upregulation of the Wound Healing Network genes at all time points, suggesting that they immediately activate late stage wound healing processes and, like the aquatic axolotl, bypass the inflammatory stage. Functional enrichment emphasized wound healing mechanisms critical for axolotl wound repair, including blood vessel development[42] (angiogenesis, proliferation/attachment of endothelial cells), tissue density[40] (permeability of blood vessel, integrity of basement membrane) and cell migration[43] (endothelial cells, fibroblasts and monocytes) in the Wound Healing Network (Fig. 4d).

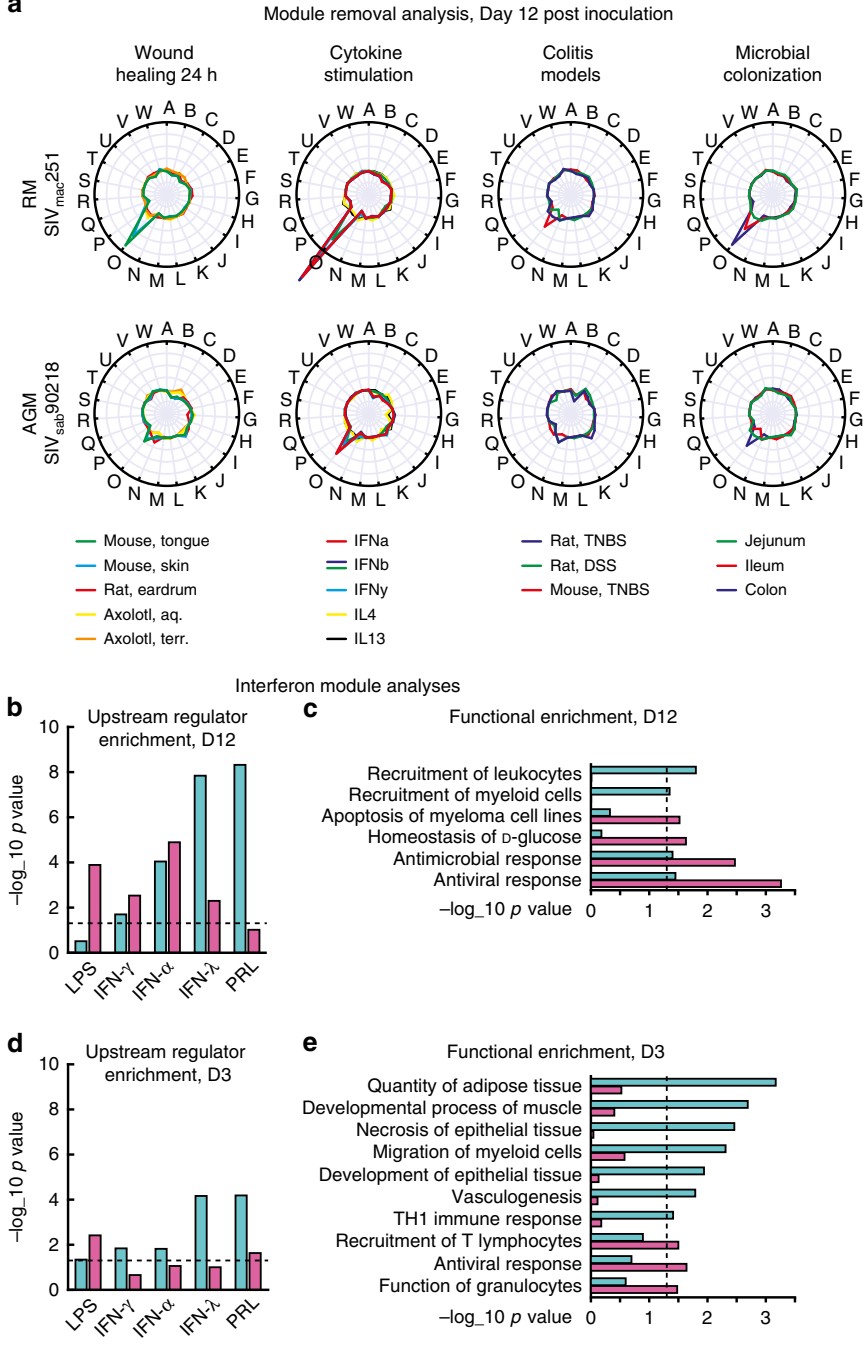

**Fig. 3** Innate immunity module analysis. **a** Module removal analysis. Radial plots show, for each module, how much its removal reduces the global correlation (Pearson correlation coefficient) between reference data and acute SIV infection at D12. Each line represents a separate reference dataset. **b**–**e** Enrichment tests of DE genes from AGMs (teal) and RMs (pink), in biological functions and upstream regulator targets, compared to all genes in the Interferon Module. Enrichment tests were performed at DE genes at D3 and D12. Dashed line indicates $p = 0.05$

**Role of fibronectin in AGM epithelial integrity preservation**. In the Wound Healing Network, the contrast between the early response to SIV infection in the AGM and RM models was similar to the contrast between early axolotl and mammalian wound repair signatures. To identify specific axolotl-like repair mechanisms in the AGM SIV model we examined which biological functions were upregulated at day 1 of axolotl wound healing by calculating the Pearson correlation coefficient between membership in each IPA biological category (0 for non-members, 1 for members), and the genes' log fold change values (Fig. 5a, Supplementary Fig. 3). Six functional categories were positively

correlated with day 1 of axolotl wound repair, including known mechanisms in this repair process (fibroblast development, infiltration of cells), but also tissue formation (adiposity, fibrous tissue development, tissue branching, colony formation).

As gene functions are frequently mediated through molecular protein interactions, we constructed a new network using protein interactions from IPA[44], between Wound Healing Network genes (Fig. 5b). The resulting network consisted mainly of ECM and cellular membrane proteins. The upregulated genes at D1 of axolotl wound repair (FN1, SDC2, FGFR1, LOXL2, and SPARC) were all associated with cell migration, and most were critical for

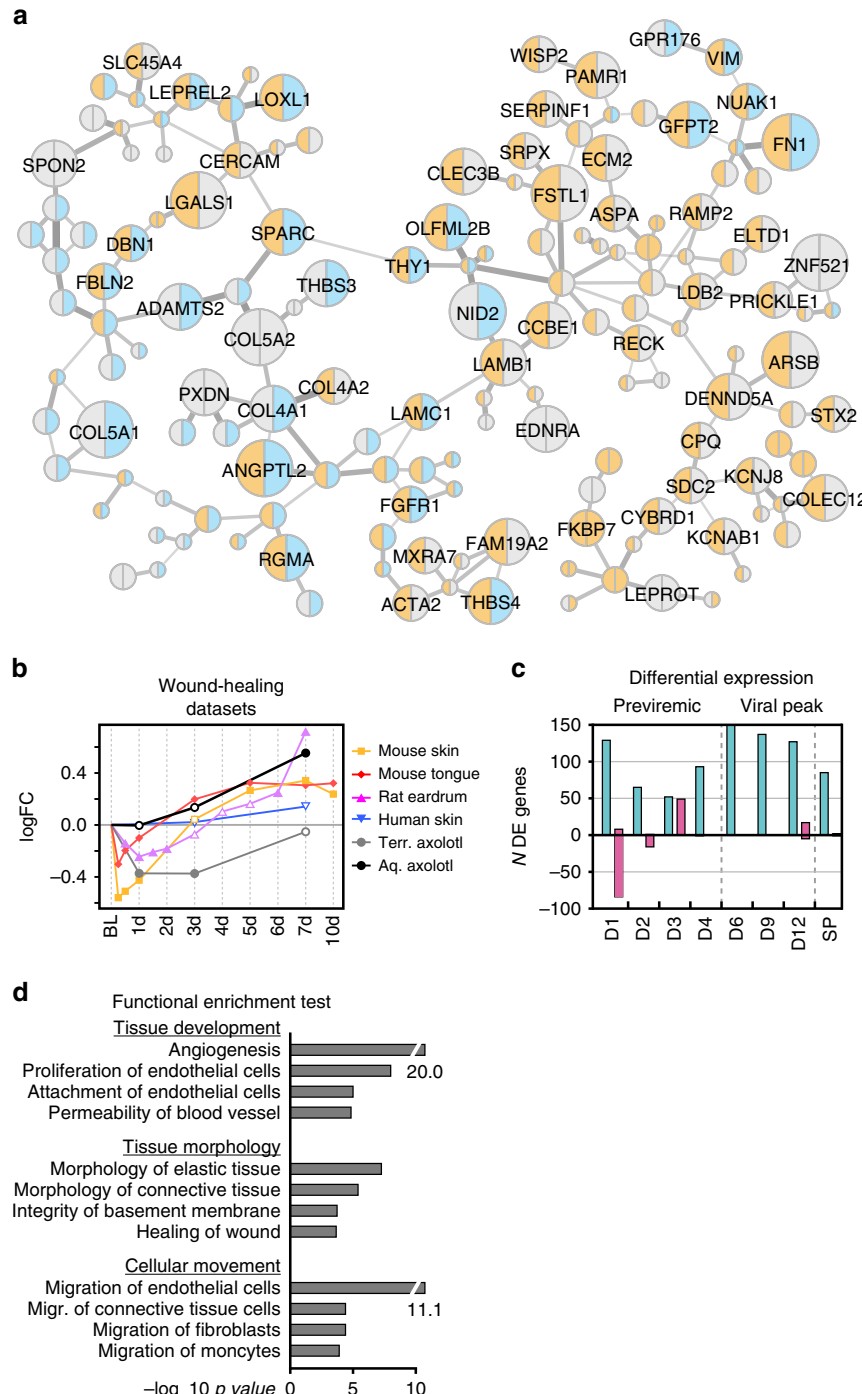

**Fig. 4** Wound Healing Network analysis revealed by CGSA. **a** Wound healing co-expression network. Node colors indicate up or downregulation (orange or blue, respectively) at D1 post inoculation in AGMs (left side of node) and RMs (right side of node). Node sizes represent the number of wound healing datasets where the gene is DE. Interaction thickness represent the weight, inherited from the Acute SIV Co-Expression Network. **b** Average log fold change values of Wound Healing Network genes from each wound healing reference dataset. Filled spots indicate significant change from baseline. Source data are provided as a Source Data file. **c** Numbers of up- and down genes in Wound Healing Network at each time point in AGM (teal) and RM (pink). **d** Functional enrichment analysis of Wound Healing Network genes, sorted into three biological themes

network connectivity. Most contacts between ECM components and cell surface proteins were mediated by FN1, through SDC2 and FGFR1, while SPARC mediated connections between fibrous collagen proteins. These observations imply that AGMs control SIV pathogenesis through an axolotl-like repair process entailing FN1 mediated recruitment of endothelial cells, fibroblasts and monocytes.

To confirm the role of the FN1 protein, fibronectin (FN), in maintenance of AGM tissue integrity, we performed immuno-histochemistry (IHC) staining in rectal tissue at baseline, D1, D3 after SIV inoculation. Quantification of FN expression showed down-regulation ($p = 0.032$, two-sided $T$-test) by day 3 in RMs, consistent with observations from the RNA-seq, albeit delayed (Fig. 5c, Supplementary Figs. 4–9). In both SIV host species, FN

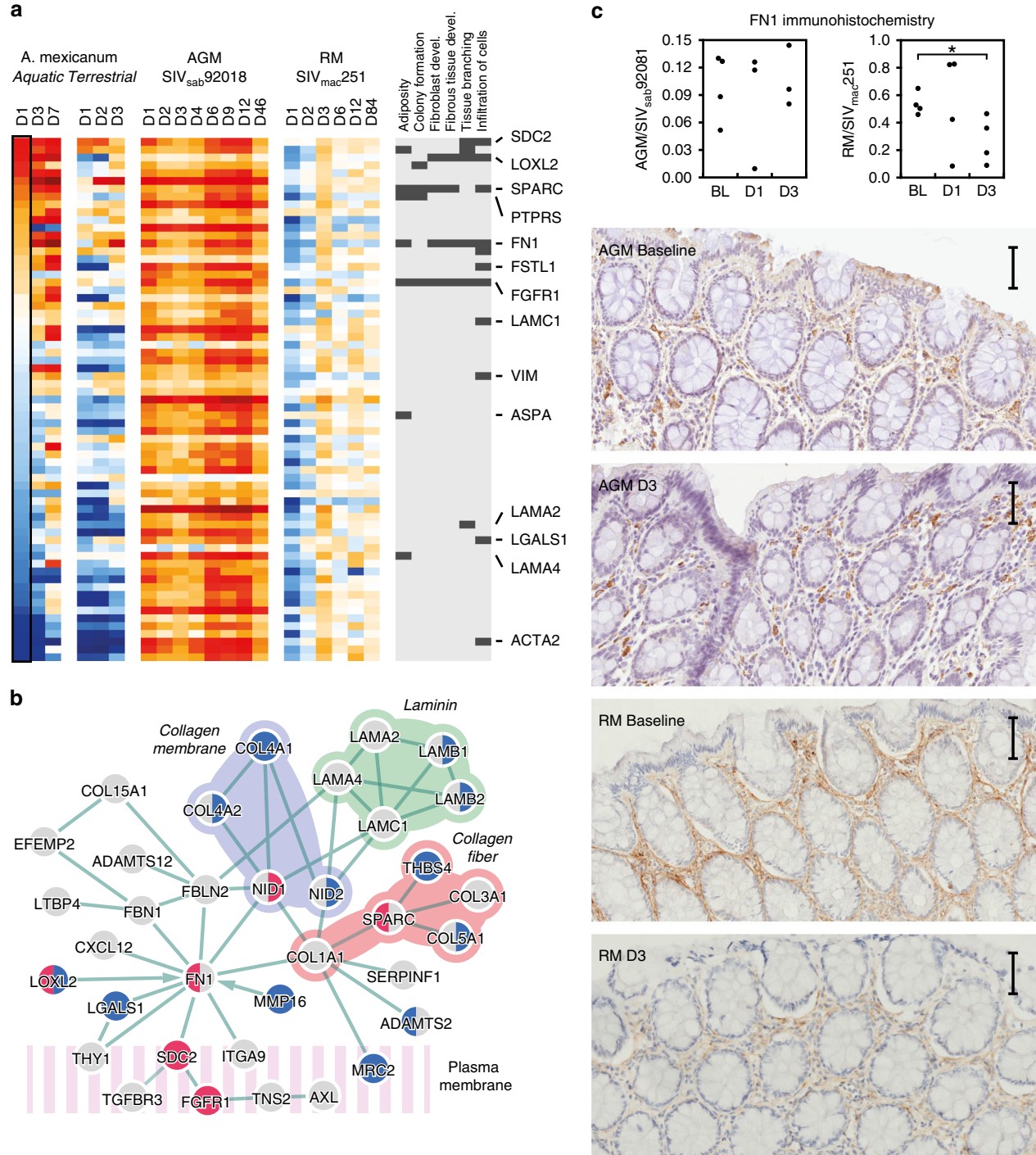

**Fig. 5** CGSA showing axolotl wound healing signatures in AGMs. **a** Heatmap of Wound Healing Network genes in axolotl wound repair and acute SIV infection, ordered by day 1 of aquatic axolotl wound repair. The biological functions that correlate with D1 axolotl wound repair, and their associated genes are shown on the right. **b** A protein interaction network between proteins encoded by genes in the Wound Healing Network. Red and blue nodes indicate up and downregulation in aquatic (left) and terrestrial (right) axolotl at day 1. **c** Immunohistochemistry staining of fibronectin (FN) in rectal mucosa at baseline, D1 and D3 post SIV inoculation. Right panels show quantification by pixel density, middle and left panels show fibronectin protein expression in lamina propria. Representative images are shown below. The scale bar included in each image represents a length of 50 μm

was primarily expressed in the lamina propria. However, in RMs FN showed more fibrillar expression.

**TGF-ß-associated wound repair signatures in AGM monocytes.** Normal wound healing requires that each growth factor, cytokine, and cell type is activated within precise time windows[33]. Monocytes

participate in both early and later stages of the process, as they can differentiate into inflammatory M1 macrophages or regulatory M2 macrophages. M2 macrophages orchestrate later stages of mammalian wound repair and are essential for axolotl wound healing[45,46].

We proceeded to investigate if macrophages support the wound healing mechanisms found in AGMs by double IF

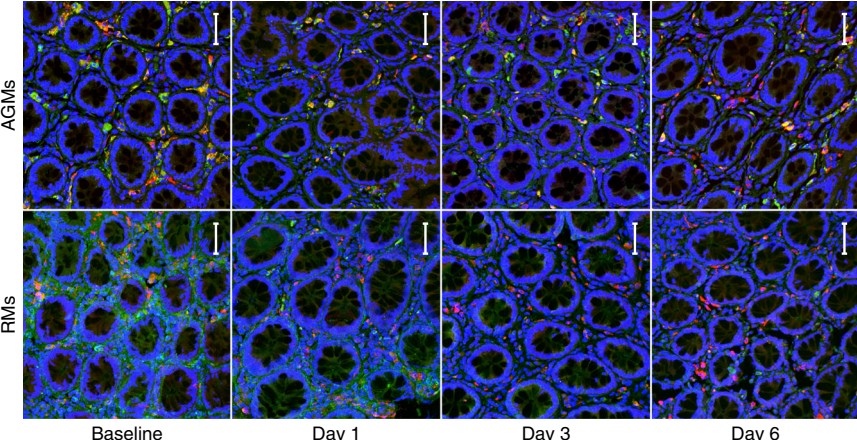

**Fig. 6** Immunofluorescent staining for HAM56/FN in lamina propria. Rectum from AGMs (top) and RMs (bottom) was doubled stained for HAM56 (red) and FN (green), with a DAPI stain (blue) to visualize nuclei. Colocalization, if any, appears as yellow. From left to right, representative images are shown from 0–6 days post-infection. All images are at ×200 magnification. Each image represents a maximum intensity projection form a z-stack of 11–28 images at 1.78 μm per step, with a resolution of 0.321 μm/pixel. The scale bar included in each image represents a length of 50 μm

staining of macrophage marker HAM56 with the wound healing protein FN (Fig. 6, Supplementary Figs. 10, 11). This showed while many HAM56+ cells were also FN+ in AGMs, most HAM56+ cells were FN− in RMs. Although single HAM56 IHC staining did not support higher numbers of macrophages in the AGMs (Supplementary Fig. 12), computational cell type deconvolution using ImmQuant[47], showed a decreased monocyte signature in RMs during the pre-viremic stage (Supplementary Fig. 13). We hypothesized that while the numbers of macrophages were similar in the two models, they exhibit a stronger wound healing function in AGMs.

To test this hypothesis, we performed deep mRNA-seq on M1 (CD14+) and M2 (CD14−/CD16+) monocytes isolated from blood samples taken at days 7, 14, and 28 post-inoculation from a separate set of animals (Supplementary Fig. 14); two AGMs and four RMs. Due to the small number of animals, we avoided DE analysis, and leveraged the genome-wide dataset by a PC analysis using the 1000 genes with the highest variance. The first principle component showed separation between AGM and RM monocytes, while the second separated M1 from M2 subtypes (Fig. 7a), including samples taken before SIV inoculation. These results show clearly that for both AGM and RM monocytes, M1 and M2 monocytes had distinct transcriptomic profiles, before and after SIV infection. The 300 genes most over-expressed in AGMs (the AGM monocyte signature) was associated with embryonic tissue development (abnormal morphology of embryonic tissue, morphology of ectoderm) and epithelial-mesenchymal transition (EMT) of epithelial cells (Fig. 7b). These gene signatures indicated that AGM monocytes have stem cell-like properties and greater differentiation potential, in keeping with axolotl-like tissues. ImmQuant also showed an increase in stem cell types after SIV infection in AGMs, including hematopoietic stem cells (Supplementary Fig. 13). In contrast, the corresponding RM monocyte signature was enriched in macrophage-mediated antiviral immunity.

As the monocytes and rectal tissues analyzed here were obtained from different sets of animals, we determined the overlap between the genes in the AGM or RM monocyte signatures to the DE genes from the rectal tissue data (where most monocytes are differentiated into tissue macrophages). In AGM rectal tissues, genes from both the AGM and RM monocyte signatures were differentially expressed throughout the course of

infection (Fig. 7c). In RMs, by contrast, both signatures are predominantly downregulated during the previremic stage of infection (D1–D4). Only at D12, the immune-associated RM signature was upregulated. Thus, AGM rectal tissues showed a wide range of monocyte responses, including both tissue development and immune activation. RM tissue macrophages, by contrast, lack the wound healing profile of AGM macrophages, leaving a disproportionately inflammatory monocyte response to SIV.

To investigate how monocytes contribute to wound healing in AGM rectal tissue, we examined the expression of predicted upstream regulators of the Wound Healing Network from IPA, in AGM and RM peripheral blood monocytes. Out of the top wound healing regulators that were expressed in the monocytes, three (TGFB1, ERBB2, and SMAD7) showed higher average expression in AGM monocytes (both M1 and M2) compared to RM. One, AHR, had comparable levels of expression in the two models (Fig. 7d).

During axolotl limb regeneration, macrophages secrete TGF-β to supply the growing tissue with new cells through influx and proliferation[48]. The remaining regulators encoded an epidermal growth factor receptor (ERBB2), nuclear receptor (AHR) and transcription factor (SMAD7), that presumably regulate gene expression in the monocytes themselves. We, therefore, examined which Wound Healing Network genes were over-expressed in AGM monocytes. Among the 1000 most variable genes in the monocyte dataset, 19 were Wound Healing Network genes (Fig. 7e). Three of the 19 genes were upregulated at D1 of axolotl wound repair: FN1, TGF-β release factor LTBP3 and cellular receptor SDC2, a mediator of TGF-β signaling[49]. The PCA analysis also included WLS, a mediator of wnt secretion, which was elevated in AGMs. Wnt is another major pathway in tissue regeneration in several species, including axolotl[50]. Taken together, these results support that monocytes/macrophages regulate AGM epithelial tissue repair during acute SIV infection, through TGF-β secretion. While wound healing is predominantly associated with M2 macrophages, we observed wound healing gene expression signatures in both classical M1 and M2 monocytes in AGMs, supporting the conclusion that the tissue repair phenotype is dynamic and directed by monocytes in general in AGMs as a feature of protection against SIV pathogenesis[18].

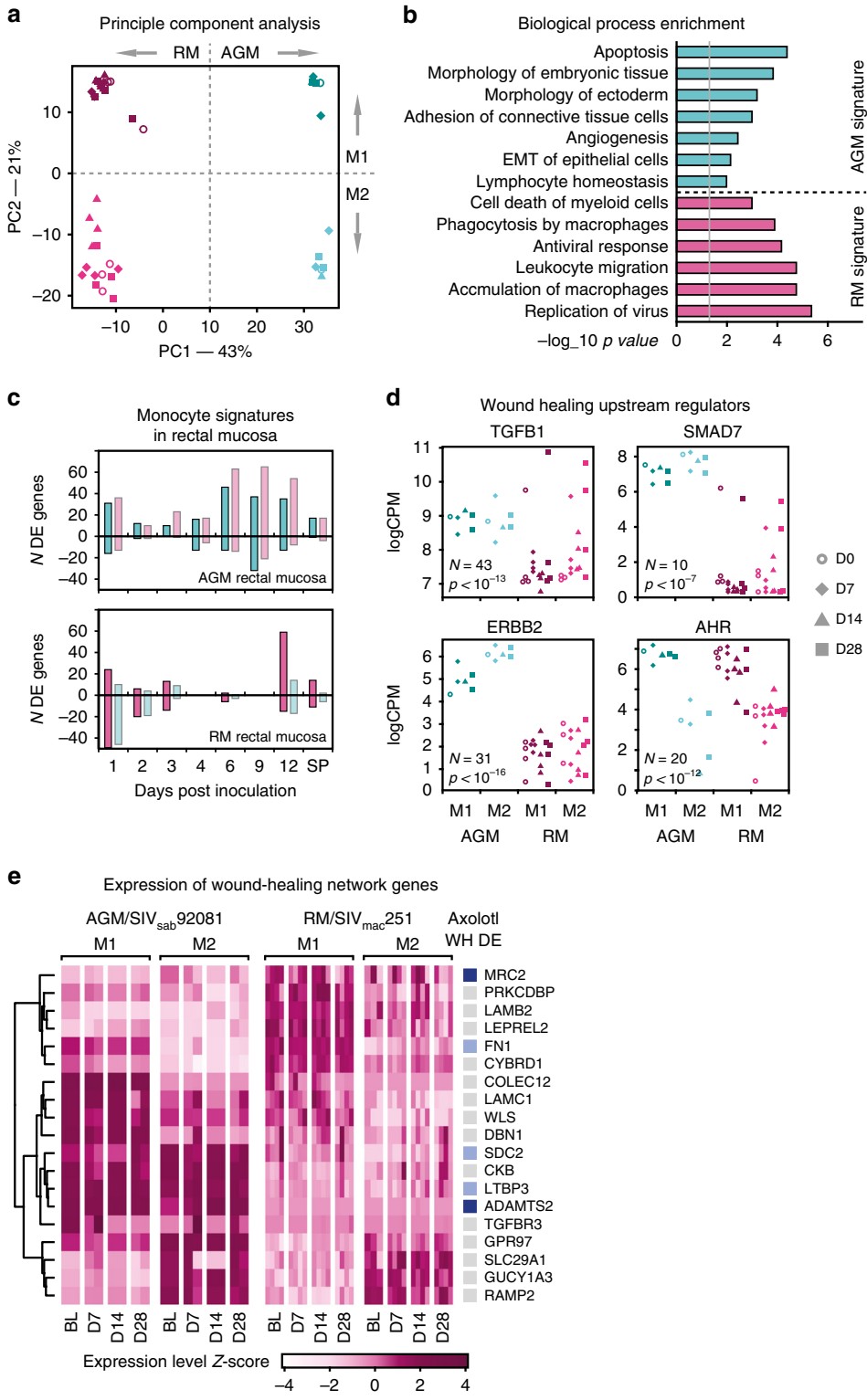

**Fig. 7** Monocyte profiling from separate set of RMs and AGMs. **a** PCA analysis of 1000 most variable genes, showing clear separation between AGM (teal) and RM (pink) monocytes, as well as type M1 (dark color) and M2 (light color). Shapes represent time points: circle—baseline, diamond—d7, triangle—d14, square—d28. **b** Functional enrichment of the AGM and RM monocyte transcriptomic signatures—each consisted of the 300 genes with the strongest positive and negative correlation to the first PC shown in (**a**). **c** Number of up- and downregulated genes from AGM and RM rectal mucosa, that overlapped with genes in the AGM or RM monocyte signatures. **d** Expression levels for predicted upstream regulators of Wound Healing Network genes in the AGM and RM monocytes from peripheral blood. Number indicate the number of regulator target genes in the Wound Healing Network, and *p*-values pertain to the corresponding enrichment test. Colors and shapes represent species, monocyte subset and time point as in (**a**). **e** Heatmap showing expression of Wound Healing Network genes in AGM and RM monocytes, with upregulated genes from axolotl wound healing. Since the most variation was introduced by species and monocyte subset, not time point, we show expression Z-score instead of logFC values. The leftmost column shows what genes are significantly upregulated in axolotl wound healing. Dark blue marks genes upregulated at day 1; light blue marks genes upregulated at day 7

## Discussion

Natural hosts of SIV avoid pathogenesis and maintain immune cell homeostasis, despite persistent viremia. Understanding how this is achieved could inform development of therapies that prevent inflammatory disease and AIDS. To determine how AGMs, a natural SIV host, control pathogenesis, we designed a study to compare AGMs with RMs, a progressive SIV model, at the site of mucosal inoculation, over an acute infection time course up to 50–85 days post-infection. This design complements previous transcriptomic studies, which have focused either on later time points, or peripheral blood or isolated immune cells[19,27].

As transcriptomic profiling of bulk tissue lacks biological resolution regarding e.g. cell types, we developed CGSA, a unique bioinformatic pipeline that integrates curated transcriptomic reference data from diverse species to interpret gene expression signatures, avoiding limitations in preexisting knowledge. The large amount of CGSA reference data also adds statistical robustness and reduces the false gene–gene associations common in co-expression analysis, as biologically relevant interactions tend to be conserved[51].

While examining the early events post mucosal SIV exposure we found stark contrasts between AGMs and RMs immediately after SIV challenge, that involved thousands of genes. A major feature in the AGM response was a unique regenerative wound healing signature activated by blood monocytes and likely propagated by tissue macrophages. This repair process was linked with preservation of mucosal integrity and absence of an inflammatory phenotype that typically associates with the tissue damage in progressive HIV/SIV hosts. Thus, AGM monocyte/macrophage signaling and the underlying wound healing signature could prime mucosal immunity for the prevention of disease progression in SIV/HIV infection. However, other immune cells are likely involved in this process and should be explored further.

AGM wound healing signature was discovered by CGSA using data from a salamander species, the axolotl. Axolotls have undifferentiated, fetal-like tissues that bypass the inflammatory healing stage. After wounding, axolotl epithelial cells and keratinocytes migrate rapidly across the wound to restore the external barrier, then initiate basement membrane regeneration and tissue remodeling[43]. This produces scar-free, fully functional new tissue[40]. Indeed, limiting tissue damage without killing the pathogen can achieve immune tolerance, avoiding inflammatory responses to microbial infections[52]. Thus, the AGM wound healing phenotype links with protection against disease progression in SIV infection.

Also, we have recently reported genetic factors that could support this wound healing profile in AGMs, including accelerated evolution in genes that control viral interactions and growth factor signaling[53], and a TLR4 variant that could limit LPS responses[54], and thus tissue damage.

The wound healing network included several constituents of the basement membrane, a collagen-rich sheet which separates the intestinal epithelium from the lamina propria and is one of the first tissue components to re-grow after limb amputation in salamanders[55]. These included collagens (major structural component), laminins (anchor cells to basement membrane[56]), nidogens (tether collagen to laminin[56]) and fibrilin (improves matrix elasticity[57]). Collagen deposition in axolotl wound healing is minimal, but other ECM proteins are transiently expressed at wound margins, including fibronectin. Fibronectin is a large ECM protein, crucial for both early and late wound healing stages[58] and expressed in the basement membrane of re-generating axolotl limbs[59]. Fibronectin matrix formation starts at cellular surfaces; $\alpha5\beta1$ integrin binds soluble fibronectin to create complexes, that are stretched into fibrils through cellular contractions. Here, IHC imaging showed that in AGM mucosal tissue the fibronectin was concentrated to cells, where polymerization is initiated during wound repair[58]. In RMs, fibronectin was predominantly present as extracellular fibrils. The cellular fibronectin expression in AGMs could accelerate wound healing compared to RMs. Both the embryonic development signature and fibronectin tissue localization were present in AGM monocytes pre-inoculation. Thus, the intestinal mucosa of AGMs, a natural SIV host, is constitutively poised to mount the protective wound healing response upon infection. Axolotl keratinocytes are similarly poised to migrate into fresh wounds, due to constitutive integrin expression[60].

Axolotl regeneration is based on their embryonic maturation state, and entails a distinct immune cell profile with low neutrophil infiltration and specific recruitment of TGF-$\beta$ secreting M2 macrophages[48]. Although mammalian wound healing is also associated with M2 macrophages, AGM monocytes of both subtypes showed a distinct embryonic tissue development profile, including the TGF-$\beta$ gene. Thus, CGSA expands the wound healing/M2 macrophage paradigm to protection from SIV disease progression and will be explored for other immune cells that regulate this process.

Therapeutic strategies to activate the wound healing monocyte/macrophage phenotype in humans could serve to suppress and control HIV disease progression in the era of antiviral therapy. Therapies such as administration of IL33 to cutaneous wounds, have shown beneficial effects in accelerating healing, enhancing re-epithelization, activating M2 macrophages and increasing fibronectin expression in mice[61]. We conclude that inducing wound repair processes is beneficial to limit SIV or HIV pathogenesis.

## Methods

**Study animals and ethical approval**. All animal experiments in this study received ethical approval by Institutional Animal Care and Use Committees (IACUCs): University of Pittsburgh IACUC for the AGM study (protocol #1008829), University of Washington IACUC for the RM study (protocol #214207) and Emory University IACUC for the monocyte study (protocol #2002173). AGMs were housed at the RIDC animal facility of the University of Pittsburgh according to regulations set forth by the Guide for the Care and Use of Laboratory Animals and the Animal Welfare Act. RMs were housed at the Washington National Primate Research Center according to guidelines approved by the University of Washington Environmental Health and Safety Committee, the Occupational Health Administration, the Primate Center Research Review Committee.

**Mucosal SIV inoculation**. Each AGM and RM was intrarectally inoculated with SIV$_{sab}$92081 and SIV$_{mac}$251, respectively, established as relatively low doses that could reliably infect its host. SIV$_{sab}$92081 naturally infects AGMs;[62] the AGM inoculum was blood plasma originally collected from an acutely infected AGM, diluted to contain $10^7$ RNA copies of SIV$_{sab}$92081. The SIV$_{mac}$251 inoculum was 1 ml containing 6000 TCID$_{50}$s (50% tissue culture infective doses).

A similar pattern of virus replication was reported in both species used in this study[11,31], although SIV$_{mac}$251 is pathogenic in RMs and SIV$_{sab}$92081 is nonpathogenic in AGMs. However, SIV$_{sab}$92081 is highly pathogenic in other primate species, including pig-tailed macaques[63] and does not inherently lack pathogenicity. Due to varying viral replication rates and pathogenicity in non-adapted SIV hosts[10,64], we compared AGMs and RMs infected with their respective adapted SIV.

To create baseline transcriptomic profiles, pinch biopsies were taken 50 days (AGMs) or 14 days (RMs) prior to inoculation. At necropsy, all rectal tissue samples were taken at a depth of 25 mm from the anus, corresponding to the depth at which the inocula were deposited. Animals were necropsied at time points that describe the pre-viremic stage (D1–D4), viral replication (D5–D12) peak and set point (D50–D84). To capture the initial host-virus interactions that determine infection outcome, about 50% of necropsies were done during the pre-viremic stage.

Blood samples were taken for viral load measurement at the same time points.

**Rectal tissue transcriptomic profiling**. Rectal tissues were immediately perfused in RNAlater and stored at −80 °C until further processing. Whole transcriptome libraries were constructed using the TruSeq Stranded Total RNA with Ribo-Zero

Gold (Illumina, San Diego, CA) as per the manufacturer's instructions. Libraries were quality controlled and quantitated using the BioAnalyzer 2100 system and qPCR (Kapa Biosystems, Woburn, MA). The resulting libraries were then sequenced initially on a HiSeq 2000 using HiSeq v3 sequencing reagents, with read number finishing using a Genome Analyzer IIx using GA v5 sequencing reagents, both of which generated paired end reads of 100 nucleotides (nt). The libraries were clonally amplified on a cluster generation station using Illumina HiSeq version 3 and GA version 4 cluster generation reagents to achieve a target density of approximately 700,000 (700 K)/mm$^2$ in a single channel of a flow cell. Image analysis, base calling, and error estimation were performed using Illumina Analysis Pipeline (version 2.8).

**Bioinformatic analyses.** Differentially expressed (DE) genes were identified using the edgeR library, and defined has having an adjusted p-value (*false discovery rate*) below 0.05 and absolute fold change above 1.5.

For CGSA co-expression network construction, log CPM values from RM/SIV$_{mac}$251 and AGM/SIV$_{sab}$92081 was combined into one matrix, creating one row for the RM and AGM homolog of each human gene. To focus the analysis on the response to SIV inoculation (and not baseline differences) the average baseline expression value for RM and AGM data was subtracted from their respective time series, resulting in log CPM fold change values (i.e., positive values where genes show above-baseline expression and vice versa).

The Acute SIV Co-expression Network was created from the pool of DE genes using rank-based network construction[25], using midweight bicorrelation (bicor)[65] as distance metric. The only parameter for this method is the number (d) of interactions to assign each gene – which we selected using GO biological process similarity[66]. We found that d values of 1 and 2 added interactions with high biological relevance, compared to when d was increased to 3 and above. We, therefore, selected the d value of 2.

Transcriptomic datasets from Gene Expression Omnibus were included based on 4 criteria: (1) Relevance for biological process in which DE genes were enriched, shown in Fig. 1d; (2) Data was obtained from relevant tissue, e.g. epithelial cells or tissue, colon biopsies based on enriched GO terms; (3) A high number of samples, for statistical power; (4) Data described a time series similar to the acute SIV infection time course, or a large clinical study. Each interaction between two genes (termed a and b) was given a weight, defined as a weighted mean between bicor values in acute SIV infection and the 23 reference datasets (equation 1).

$$W_{a,b} = \left(A_{a,b} * 1 + R_{a,b} * 5.75\right)/(1 + 5.75) \quad (1)$$

Where $A_{a,b}$ is the bicor value between gene a and b in the acute infection data; $R_{a,b}$ is the bicor provided by the reference datasets. In reference datasets from mouse, rat and axolotl, probe IDs were mapped to ensemble gene ID, and to their human ortholog where one-to-one orthologs were available. Otherwise, they were excluded from the rest of the analysis. The weight (5.75) was selected to optimize GO term similarity values in the resulting network.

$$R_{a,b} = \left(\sum\nolimits^{23}_i = 1 D_{a,b,i} * \mu_i\right)/\left(\sum\nolimits^{23}_i = 1 \mu_i\right) \quad (2)$$

Where $D_{a,b,i}$ is the bicor value between gene a and gene b in reference dataset i. Given that the datasets can be of varying size and quality, the datasets were given individual weights (u), defined as the Spearman correlation coefficient between the vector of bicor values for the dataset (D) and corresponding GO term similarity values. Biological modules were identified in the network using a modularity optimization algorithm[67].

**Immunohistochemistry.** Rectal tissues from D1, D3, and unchallenged animals, were dissected from euthanized monkeys and fixed in SafeFix II (RMs) (cat# 23-042-600, Fisher Scientific) or paraformaldehyde (AGMs) and embedded in paraffin. Antibodies against FN (1:5000 dilution; cat# 3776-1, Epitomics Inc.) and HAM56 (1:100 dilution, cat# 14-6548-93, eBioscience) were used for IHC by procedures described previously[29]. FN IHC was performed by Q.L. and W.L. The HAM56 IHC was performed by I.P. and K.R.

The FN stained slides were digitized using ScanScope (Aperio, Vista, CA). For quantitative image analysis, the images of slides were imported into ImageScope; regions of lamina propria and epithelium were selected using ImageScope drawing tools. FN expression was quantified using a positive pixel count algorithm in the Spectrum Plus analysis program (version 9.1). The parameters of the algorithm were manually tuned to accurately match the FN+ markup image over the background DAB stain. Once the parameters were set, the algorithm was applied automatically to all images to measure FN+ signals by area.

Quantitative image analysis was also performed for HAM56 using methodology similar to Somsouk et al.[68]. Briefly, slides were imaged using an AxioCam MRc5 (Zeiss, Oberkochen, Germany) and Axiovision software v.4.7 (Zeiss). Regions of the rectal lamina propria to be imaged were selected at random to try to minimize bias. Lymphoid aggregates were avoided, as they are structurally and immunologically distinct from the lamina propria. All images were taken at either ×100 or ×200 magnification, depending on the quantification technique. After image collection, multiple images from the same

animal were processed by applying the Color Deconvolution function in the FIJI image processing package (version 2.0)[69] to isolate the DAB coloration. An intensity threshold was then manually applied to each image to select the positive DAB. The total percent area occupied by the positive DAB signal was then measured and averaged for all the images from the same animal.

We histologically examined fibronectin expression by macrophages in the gut by performing a double immunofluorescence stain for FN and HAM56, as described previously[70]. Briefly, the paraffin tissue sections were deparaffinized by a battery of three 5 min washes in xylene and then rehydrated by 3–5 min sequential washes in 100, 95, and 75% ethanol. Next, the slides were boiled in diluted Antigen Unmasking buffer (Vector Laboratories, Burlingame, CA) for 20–23 min before being allowed to cool to room temperature. The slides were then washed three times in 1× PBS for 5 min. The slides were then blocked with Protein Block (Dako, Santa Clara, CA) for 30 min. The blocking solution was dumped off and the slides were incubated with the FN antibody (mouse IgG, ThermoFisher, clone FBN11, ref# MA5-11981) diluted to 1:100 with Antibody Diluent (Dako, Santa Clara, CA) for 1 h at room temperature. After incubation with the first primary antibody, the slides were again washed three times in 1× PBS before incubating with the corresponding secondary antibody (donkey antimouse IgG, Abcam, Alexa Fluor® 488, ref# ab150061) diluted to 1:100 with Antibody Diluent for 30 min. The slides were then washed three times in 1XPBS and incubated with the HAM56 antibody (mouse IgM, ThermoFisher, clone HAM56, ref# 14–6548) diluted to 1:100 with Antibody Diluent for another 60 min. Following another wash three times in 1× PBS, the slides were incubated with the corresponding secondary antibody (goat antimouse IgM, ThermoFisher, polyclonal, Alexa Fluor® 633, ref# A-21046) diluted to 1:100 with Antibody Diluent for 30 min. The slides were then washed more times in 1× PBS and incubated for 5 min with DAPI (MilliporeSigma, Burlington, MA) at 1:5000 dilution in 1× PBS and then washed again. To reduce lipofuscin autofluorescence, the slides were incubated for 30–60 s with TrueBlack autofluorescence quencher (Biotium, Fremont, CA) and then washed in 1× PBS. Finally, the slides were washed for 5 min in 1× PBS and then coverslipped using Fluorescence Mounting Medium (Dako). After staining, the slides were stored in the dark at 4 °C until they were visualized with an Olympus Fluoview 1000 Confocal microscope. Maximum intensity projections (IPs) were generated from image z-stacks of 11–28 images using NIS Elements 5.20.00 (https://www.lim.cz/). The maximum IPs were processed and edited using FIJI version 2.0[69] (https://fiji.sc/).

**Monocyte sorting and RNA sequencing.** AGM and RM PBMCs were isolated from EDTA-treated whole blood with dilution in sterile phosphate-buffered saline (PBS) and centrifugation for 30 min at 1850 RPM at 25 °C in Ficoll-Paque™ (Lonza) at a 3:2 ratio. The isolated buffy coat was washed with PBS and contaminating red blood cells lysed using an ammonium-chloride-potassium lysing buffer (Lonza) for 10 min before washing with PBS (10 min, 1800 RPM, 25 °C). Cells were counted using LIVE/DEAD® Aqua Dead Cell Stain Kit (Life Technologies), surface-stained with antibodies against CD14 and CD16 (Beckman Coulter) and incubated for 30 min at RT before washing with FACS buffer (PBS + 2% FBS) (Supplementary Fig. 14). Samples were sorted using a FACS ARIA II (BD Immunocytometry) and analyzed using FlowJo software. Collected monocytes (~50,000 cells/sample) were stored in RLT. RNA was isolated using RNeasy Micro Kit (Qiagen,) including DNase Digestion (Qiagen, 74004 and 79254). RNA quality and quantity were determined using a Bioanalyzer RNA Pico chip (Agilent, 5067-1513). The SMARTer Ultra Low Input RNA Kit for Sequencing—v3 (Clontech/TAKARA, 634851) was used for first-strand cDNA synthesis and cDNA amplification (14 cycles) starting with ~2 ng of total RNA. The amplified cDNA was cleaned using DNA Clean & Concentrator™-5 kit (Zymo, D4014) and quantified using Qubit dsDNA High Sensitivity kit (Thermo, Q32854). Using the resulting purified cDNA (500 pg/library), the library construction was completed using three steps in the Illumina's Nextera XT Library Prep (Illumina, FC-131-1096) workflow (Tagment DNA, Amplify Tagmented DNA, and Clean Up Amplified DNA). The resulting libraries were quantified (Qubit dsDNA High Sensitivity Kit) and sizes confirmed using a Bioanalyzer High Sensitivity DNA Chip Kit (Agilent, 5067–4626). Libraries were sequenced to a depth of ~25 million raw reads/sample using a NextSeq 500, and 150 cycle, NextSeq 500/550 High Output v2 kits (Illumina, FC-404-2002).

**Reporting summary.** Further information on research design is available in the Nature Research Reporting Summary linked to this article.

## Data availability

The RNA sequencing data from both rectal tissues and monocytes can be found in Gene Expression Omnibus under accession number GSE111234. Data underlying Fig. 4b are provided as a Source Data file. All other data are available from the corresponding author upon reasonable request.

## Code availability

The complete R code is available upon reasonable request (confirmed compatible with R version 3.5.0).

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

## Acknowledgements

We would like to thank Michael Katze and Robert Palermo for their input during the early stages of this study. This work was supported by the National Institutes of Health, Office of the Director P51OD010425, R24OD011157, and R24OD011172; University of Washington Center for Innate Immunity and Immune Disease; NIAID contract no. HHSN272201300010C; an NIAID Simian Vaccine Evaluation Unit contract with the University of Washington (contract no. N01- AI-60006); NIDDK, NCRR and NHLBI, NIH (R01 DK087625-01. to Q.L., R24-OD010445 to S.E.B. and G.S., RR025781, DK108837, R01HL117715, R01HL123096, R01DK113919, R01AI119346 to I.P. and C.A.); the Preclinical Research & Development Branch, VRP, DAIDS, NIAID, NIH (task order under N01-AI-30018 to Q.L.); the DAIDS Reagent Resource Support Program for AIDS Vaccine Development, Quality Biological, Gaithersburg, MD, Division of AIDS (contract no. N01-A30018); National Institutes of Health Training Grant T32 AI065380-08 and AI065380-09 (to K.R.) and the Swedish Research Council (D0045701) and eSSENCE (to F.B.).

## Author contributions

This study was conceived and designed by: I.P., C.A. and M.G. Experiments were performed by: K.R., C.X., S.E.B., W.L., Q.L., J.Z., J.C., E.S., J.M.W., R.A.D., G.H.R., A.T. and D.M. Transcriptomic data were acquired by: R.R.G., A.N., M.J.T. and X.P. Materials were contributed by: S.E.B. and G.S. Bioinformatic analyses were performed by: F.B., R.R.G., and J.K. Results were analyzed and interpreted by: F.B., K.R., L.L., S.E.B., C.A., I.P. and M.G. The paper was written by: F.B., K.R., R.R.G., L.L., I.P. and M.G. I.P. and M.G. contributed equally to this study.

## Competing interests

The authors declare no competing interests.

## Additional information

**Peer Review Information** *Nature Communications* thanks Marina Sirota and the other, anonymous, reviewer(s) for their contribution to the peer review of this work. Peer reviewer reports are available.

