## [Peer Review File · Nature Communications]

Reviewers' comments:

Reviewer #1 (Remarks to the Author):

This is a very interesting and well-written paper from Gale and colleagues that helps considerably to advance our understanding of non-pathogenic SIV infection in natural hosts. The experiments are well performed and the analysis is very rigorous.

My main concern is that the conclusions are somewhat overstated given the data discussed. I think everything could be toned down and placed in some context. This begins with the title which reads as if the authors have identified THE mechanism underlying non-pathogenic infection. It certainly might be A mechanism that contributes but not the only one. After all, the observations (from Brenchley) of CD4 downregulation in memory T cells to reduce target cell availability have to play some role too.

The identification of macrophage-associated wound healing pathways is very interesting but it biases the reader into regarding the entire process as mediated by macrophages which is unlikely. These were the cells that were analyzed and so it's no surprise that these are the pathways identified.

Finally, it would help considerably if the hypothesis were actually tested in SIV-uninfected AGM and RM. This shouldn't be too difficult to accomplish. One could examine recovery in different situations such as (1) DSS administration, (2) IFN α administration and (3) after skin punch (see Keyes et al, Cell 2016 and Linehan et al, Cell 2018).

Reviewer #2 (Remarks to the Author):

The study by Barrenas, Gale and colleagues addresses the following question: what is the underlying mechanism by which African Green Monkeys naturally infected with SIV_{agm} avoid the pathogenic effects of HIV/SIV infection typically observed in humans and rhesus macaques. It has long been appreciated that naturally infected AGMs and Sooty Mangabeys show minimal signs of AIDS despite high levels of viremia and depletion of gut CD4⁺ T cells. There is a large body of literature that addresses this subject insofar as understanding how these two primate species have evolved to tolerate SIV infection might provide important insights into the underlying mechanisms that promote HIV disease in humans. Previously published studies have noted a lack of bacterial

translocation in Sooty Mangabeys, which raises the more specific question as to how these naturally infected primates are able to minimize damage to the integrity of gut tissues. Although not an expert in the field of NHP infection and disease, I have long regarded the particular subject of natural infection and the absence of disease progression to be a valuable area of AIDS research that holds real potential to further our understanding of AIDS pathogenesis. It is in this context that I have evaluated this manuscript.

The authors develop and employ a novel gene expression systems biology tool, termed CGSA to differentiate the expression signatures of AGMs vs. Rhesus macaques during the acute phase of infection. Their rationale is that the critical events that allow AGMs to avoid disease must occur around the time of acute infection. Critically important to this study was the analysis of expression in gut tissues, which is unlike previous studies which employed samples from peripheral blood. In addition to comparing gene expression in these primate species the authors incorporate relevant public domain databases that included a wide range of animals. Finally, to validate one of their chief leads (fibronectin) they carry out *in situ* hybridization of gut lamina propria.

Overall, I found the study to be well described, and the results and conclusions to be interesting. The identification of key roles for monocyte/macrophage -derived TGF β and fibronectin fits very well with earlier reports involving the mechanisms underlying SIV/HIV mediated damage to gut tissues. The systems biology approach that the authors adopted is advanced and the strength of the study. It was well-described and I was able to appreciate the findings. The experimental design was sound. However, the validation of the leads developed by this analysis was less extensive.

Major comments:

1. More information should be added in the text about the choice of the reference dataset used to perform CGSA on a whole genome scale and shown in Fig 2D. How do these reference datasets relate to the 23 reference datasets used for the CGSA on the 23 network modules of the previous panel?

2. How do the authors explain the lack of correlation between the transcriptional profile of the AGM samples and the wound healing 24h reference dataset of the aquatic Axolotl in Fig 2D (and weak correlation with Wound healing 7d)? How do they reconcile their intriguing hypothesis that the response of AGMs to SIV infection is similar to the aquatic Axolotl wound healing mechanism in absence of any correlation between the aquatic Axolotl reference dataset and the AGM transcription profile?

3. The FN1 gene was identified through an analysis of the biological functions that were up regulated at day 1 of axolotl wound healing and the assumption that the contrast between the early response to SIV infection in the AGM and RM models was similar to the contrast between early axolotl and mammalian wound repair signatures. Thus, the lack of correlation between genes in the aquatic Axolotl reference dataset and the genes upregulated by AGM following SIV infection is an important point to address. A more direct comparison of the gene signature of axolotl wound healing and SIV infection in AGM might help one understand the choice of FN1 as the gene/protein to focus on to validate the approach used by the authors.

4. In the FN1 IHC analysis, were the baseline data from the same 4 animals stained at day 3? Was the same animal used in the images shown? It is likely that the authors collected tissue blocks at baseline from all animals. To have a better idea of the level of FN1 at baseline, it would be good to the staining of additional animals.

5. The authors show colocalization of HAM56 and FN in AGMs (infected or not?). A similar staining should be carried out in RMs to understand if this colocalization is unique to AGMs would be helpful.

6. In the last paragraph of the results the authors conclude that their results support that monocytes/macrophages regulate AGM epithelial tissue repair during acute SIV infection, through TGF- β secretion. Was TGFB1 among the genes upregulated in AGMs by SIV infection in the RNAseq analysis? It would be important to confirm these findings through IHC/IF of the tissues collected at necropsy and compare AGM with RMs.

7 No description of statistical analysis was provided figure 5C is provided.

Minor comments:

Line 89 page 5: A more correct ref is: Pandrea I JI 2008.

Reviewer #3 (Remarks to the Author):

In this manuscript, Authors report that African green monkeys, natural hosts of SIV, activate wound healing mechanisms in mucosal tissues and attribute this to the ability of AGM to prevent the

development of AIDS. In contrast, SIV infection leads to the development of AIDS in rhesus macaques and these animals lack the activation of wound healing mechanisms to the same extent. Authors obtained rectal biopsies or necropsy tissue from AGM and RM animals during acute SIV infection and performed RNA-seq analysis to determine the gene expression profiles. Conserved gene signature analysis was applied to compare these data with previously published transcriptomic data sets across different species. Authors conclude that the activation of wound healing process in AGM preserves mucosal integrity, prevents inflammation mediated damage and halts disease progression. In summary, this manuscript reports a comprehensive comparative analysis of transcriptomic profiles of rectal tissues from AGM and RM during acute SIV infection and identifies gene signatures of protection from disease progression. The AGM gene data sets associated with the wound healing/repair and mucosal protection against virally induced damage provide a reference gene list and this will be of interest to HIV researchers.

There are several comments/questions that need to be addressed.

Viral load data are presented in plasma samples from AGM and RM animals. However, the gene expression data were obtained from rectal biopsies/necropsy tissue. What were the viral loads in rectal tissues? Were the levels of viral RNA similar or different in the rectal tissues from these animals?

AGM animals were infected with SIV_{sab} and rhesus macaques were infected with SIV_{mac251}. Authors should comment on the pathogenicity of these viruses and discuss the similarities/differences between these two viruses.

Previously published papers (by the Authors and other investigators) have reported on gene expression profiles in mucosal tissues of AGM and RM animals and changes during SIV infection. In the current manuscript, Authors have compared the gene expression profiles across several species based on the published data sets. In this regard, Authors have expanded the depth of the gene expression analysis and have identified the wound healing process as an important mechanism. This is definitely a comprehensive rectal mucosal gene expression analysis of AGM and RM responses to acute SIV infection. By defining the previremic, viremic and set point stages, Authors have increased the number of animals per each group and this approach has strengthened the data analysis.

It is not clear whether rectal tissue samples were longitudinally from the same animal overtime. If they are not from the same animal, what is the magnitude of variability in the spread of SIV infection in the rectal tissue of these animals? What kind of histopathologic changes are seen between AGM and RM animals? Although the general trend in the gene expression changes may be similar, an animal-to animal variation is evident in the Figure 1D.

Conserved gene signature analysis was applied to compare the acute SIV infection gene expression data sets from AGM and RM to the available transcriptomic data sets across different species and inflammatory conditions and to the pathways in axolotl wound repair. This is an interesting approach. However, there have been several papers published on the gene expression analysis in AGM and RM and in HIV infection on the recovery during ART and in RM models with vaccine protection. It would have been helpful to compare the acute SIV gene network and wound repair

genes in rectal tissue of AGM with non-progressive asymptomatic HIV or SIV infections and from vaccine protected animals.

Authors found a wound healing signature in the AGM model that included monocyte regulated mucosal tissue immune remodeling. AGM rectal tissue had increased expression of myeloid cell recruitment associated gene expression (Fig. 3E). Does histopathologic analysis substantiate this observation by increased number of macrophages in the rectal tissue of the AGM compared to RM? In Figure 6A, Ham56+ cells are shown in AGM rectal tissue. What about the RM rectal tissue samples? Are there any differences between these samples?

In the Figure 5C, the FN immunostaining in RM rectal tissue is difficult to compare with the AGM tissue. Comparable levels of nuclear stain and better images may help validate the differences.

Reviewer #4 (Remarks to the Author):

The authors aim to examine how natural hosts of simian immunodeficiency virus (SIV) avoid AIDS despite lifelong infection by comparing a natural SIVhost, African green monkey (AGM) to an AIDS susceptible species, rhesus macaque (RM) on the transcriptomic level. The authors developed a systems biology approach, referred to as conserved gene signature analysis (CGSA), based on correlating expression signatures in acute SIV infection across informative transcriptomic datasets from the public domain across diverse species. They find that during acute SIV infection, AGMs rapidly activate evolutionarily-conserved regenerative wound healing mechanisms (fibronectin) in mucosal tissue and that AGM monocytes exhibited an embryonic development and repair/regeneration signature featuring TGF- β and concomitant reduced expression of inflammatory genes compared to RMs

Since my background is in bioinformatics, I will only comment on the computational merits of this manuscript. While the findings seem interesting and biologically relevant, several points must be addressed on the analytical side:

- It seems that the authors are comparing a rectal samples after necropsy to blood samples, which is problematic. Maybe I'm missing something, but from the results section, it is unclear how the different tissue types are handled.
- From the description of the results, it is unclear how many samples per group and what method was used for differential expression. Which cutoffs were applied for significance?

- Was multiple hypothesis testing applied? The authors carry out many comparisons and this needs to be taken into account.
- Instead of the pairwise comparisons of differential expression, I would suggest looking at a time series analysis (longitudinal comparison) using linear mixed models.
- The public datasets that were chosen to be included in CGSA are based on the original comparison and pathway analysis, which is rather biased. The logic behind this is circular and the results are expected.
- The authors should explain how they are integrating data from microarrays and RNA-Seq and also across different organisms.
- Cell type deconvolution approaches could be helpful in further examining the role of monocytes.
- The limitations of the presented approaches should be listed in the discussion.

Reviewers' comments:

Reviewer #1 (Remarks to the Author):

This is a very interesting and well-written paper from Gale and colleagues that helps considerably to advance our understanding of non-pathogenic SIV infection in natural hosts. The experiments are well performed and the analysis is very rigorous.

1.1 My main concern is that the conclusions are somewhat overstated given the data discussed. I think everything could be toned down and placed in some context. This begins with the title which reads as if the authors have identified THE mechanism underlying non-pathogenic infection. It certainly might be A mechanism that contributes but not the only one. After all, the observations (from Brenchley) of CD4 downregulation in memory T cells to reduce target cell availability have to play some role too.

REPLY: While we have avoided categorical statements in our conclusions, we realize that both the title and abstract were indeed overstated. We have adjusted the text accordingly.

1.2 The identification of macrophage-associated wound healing pathways is very interesting but it biases the reader into regarding the entire process as mediated by macrophages which is unlikely. These were the cells that were analyzed and so it's no surprise that these are the pathways identified.

REPLY: This is a very important point – we have emphasized the role of other immune cells in the revised discussion.

1.3 Finally, it would help considerably if the hypothesis were actually tested in SIV-uninfected AGM and RM. This shouldn't be too difficult to accomplish. One could examine recovery in different situations such as (1) DSS administration, (2) IFN α administration and (3) after skin punch (see Keyes et al, Cell 2016 and Linehan et al, Cell 2018).

REPLY: We must respectfully disagree with the reviewer on this point. Further experiments that interfere with living primates is a massive undertaking, requiring an ethical approval process, extensive resources and significant time investment. It also needs to coincide with available animals. However, this could be a highly compelling, separate follow-up study.

We have, however, performed further histological staining of macrophage markers in the animals already sacrificed. While we did not find a difference in total macrophage numbers by HAM56 staining, by HAM56/FN co-staining, we observed a higher number of cells positive for both HAM56 and FN in AGMs, consistent with macrophages having

a stronger wound healing function in this species.

Reviewer #2 (Remarks to the Author):

The study by Barrenas, Gale and colleagues addresses the following question: what is the underlying mechanism by which African Green Monkeys naturally infected with SIV_{agm} avoid the pathogenic effects of HIV/SIV infection typically observed in humans and rhesus macaques. It has long been appreciated that naturally infected AGMs and Sooty Mangabeys show minimal signs of AIDS despite high levels of viremia and depletion of gut CD4⁺ T cells. There is a large body of literature that addresses this subject insofar as understanding how these two primate species have evolved to tolerate SIV infection might provide important insights into the underlying mechanisms that promote HIV disease in humans. Previously published studies have noted a lack of bacterial translocation in Sooty Mangabeys, which raises the more specific question as to how these naturally infected primates are able to minimize damage to the integrity of gut tissues. Although not an expert in the field of NHP infection and disease, I have long regarded the particular subject of natural infection and the absence of disease progression to be a valuable area of AIDS research that holds real potential to further our understanding of AIDS pathogenesis. It is in this context that I have evaluated this manuscript.

The authors develop and employ a novel gene expression systems biology tool, termed CGSA to differentiate the expression signatures of AGMS vs. Rhesus macaques during the acute phase of infection. Their rationale is that the critical events that allow AGMs to avoid disease must occur around the time of acute infection. Critically important to this study was the analysis of expression in gut tissues, which is unlike previous studies which employed samples from peripheral blood. In addition to comparing gene expression in these primate species the authors incorporate relevant public domain databases that included a wide range of animals. Finally, to validate one of their chief leads (fibronectin) they carry out *in situ* hybridization of gut lamina propria.

Overall, I found the study to be well described, and the results and conclusions to be interesting. The identification of key roles for monocyte/macrophage -derived TGFβ and fibronectin fits very well with earlier reports involving the mechanisms underlying SIV/HIV mediated damage to gut tissues. The systems biology approach that the authors adopted is advanced and the strength of the study. It was well-described and I was able to appreciate the findings. The experimental design was sound. However, the validation of the leads developed by this analysis was less extensive.

Major comments:

2.1. More information should be added in the text about the choice of the reference dataset used to perform CGSA on a whole genome scale and shown in Fig 2D. How do

these reference datasets relate to the 23 reference datasets used for the CGSA on the 23 network modules of the previous panel?

REPLY: The reference datasets were selected based on the biological processes identified by GO enrichment. We also required that included datasets had a high number of samples, and, where applicable, time points that matched our acute infection design. We have expanded upon this point in the revised Methods.

Furthermore, we have made an addition to the supplement assessing the contribution of each type of reference dataset to each module. Briefly, for each interaction in the global co-expression network, we identified the reference dataset where this interaction had the strongest correlation coefficient. Within each module, we counted the number of interactions that pertain to each dataset type (wound healing, colitis models, cytokine stimulation, homeostasis, cancer and IBDs).

This produced an overview of which dataset type that produced each gene-gene interaction, and by extension, gene module. The resulting figure confirms observations that were reported in the paper; that the modules most strongly correlated in wound healing datasets were also up-regulated in AGMs (left side of the network). It also showed that the module with the strongest correlations from cytokine stimulation studies was module O, referred to as the “interferon module” in the paper.

2.2. How do the authors explain the lack of correlation between the transcriptional profile of the AGM samples and the wound healing 24h reference dataset of the aquatic Axolotl in Fig 2D (and weak correlation with Wound healing 7d)? How do they reconcile their intriguing hypothesis that the response of AGMs to SIV infection is similar to the aquatic Axolotl wound healing mechanism in absence of any correlation between the aquatic Axolotl reference dataset and the AGM transcription profile?

REPLY: Indeed, the correlation between AGM acute infection, and 24h aquatic axolotl wound healing is near zero (figure 2D, upper left corner). However, this should be compared to the correlation between AGM acute infection, and inflammatory, mammalian wound healing, which is strongly negative. Thus, the near zero correlation between aquatic axolotl and AGM acute infection is substantially higher.

2.3. The FN1 gene was identified through an analysis of the biological functions that were up regulated at day 1 of axolotl wound healing and the assumption that the contrast between the early response to SIV infection in the AGM and RM models was similar to the contrast between early axolotl and mammalian wound repair signatures. Thus, the lack of correlation between genes in the aquatic Axolotl reference dataset and the genes upregulated by AGM following SIV infection is an important point to address. A more direct comparison of the gene signature of axolotl wound healing and SIV infection in AGM might help one understand the choice of FN1 as the gene/protein to focus on to validate the approach used by the authors.

REPLY: To expand upon this analysis, we have now also approached this issue from the opposite direction: we started with the up regulated genes from D1 axolotl wound healing and examined the same signature in acute SIV infection (Online supplement). This analysis identified five gene clusters which captured different aspects of non-inflammatory wound healing. Three of them were also activated in the AGM SIV model. FN1 was nested within one of the larger clusters, which was associated with collagen fibril organization, chondrocyte differentiation, muscle development and mesodermal differentiation. FN1 plays documented roles in all these processes, and can interact with many other proteins. This suggested that FN1 was important for this process, and was an appropriate candidate for further analyses.

The new figure has been included in the online supplement.

2.4. In the FN1 IHC analysis, were the baseline data from the same 4 animals stained at day 3? Was the same animal used in the images shown? It is likely that the authors collected tissue blocks at baseline from all animals. To have a better idea of the level of FN1 at baseline, it would be good to the staining of additional animals.

REPLY: The tissues used for IHC were collected upon necropsy. The “baseline” samples in this analysis were in fact collected from separate, uninfected animals as the small pinch biopsy collected as baseline pinch biopsies were not suitable for his analysis. We have added further images to the supplement.

2.5. The authors show colocalization of HAM56 and FN in AGMs (infected or not?). A similar staining should be carried out in RMs to understand if this colocalization is unique to AGMs would be helpful.

REPLY: In the revised manuscript, we have added post-infection co-staining of HAM56 and FN1 in both AGMs and RMs before infection, D1, D3 and D6. This analysis showed substantially higher frequency of FN1 positive macrophages in AGMs, in both uninfected and infected animals. However, single HAM56 staining did not support higher total quantity of macrophages. Taken together with the transcriptomic analysis monocyte, this indicates different macrophage phenotypes, but not quantities, between the two SIV hosts. These results can be found in the new/revised figures 6 and 7 as well as supporting figure S11-S13.

2.6. In the last paragraph of the results the authors conclude that their results support that monocytes/macrophages regulate AGM epithelial tissue repair during acute SIV infection, through TGF- β secretion. Was TGFB1 among the genes upregulated in AGMs by SIV infection in the RNAseq analysis? It would be important to confirm these findings through IHC/IF of the tissues collected at necropsy and compare AGM with RMs.

REPLY: TGFBI was one of many genes that were significantly down regulated in RMs within days of infection, but did not change significantly in AGMs. It should also be noted that across all samples, including baseline, TGFBI expression was higher in AGMs than RMs; mean (SD) of normalized counts were 198.5 (79.7) and 116.1 (92.5), respectively.

However, TGFBI is known to be difficult to detect by IHC, as it's secreted in an inactive form and its expression level may not represent its activity. We therefore chose to focus additional analyses on the macrophages themselves.

2.7 No description of statistical analysis was provided figure 5C is provided.

REPLY: We have clarified that this was done by a t-test.

Minor comments:

Line 89 page 5: A more correct ref is: Pandrea I JI 2008.

This reference has been replaced.

Reviewer #3 (Remarks to the Author):

In this manuscript, Authors report that African green monkeys, natural hosts of SIV, activate wound healing mechanisms in mucosal tissues and attribute this to the ability of AGM to prevent the development of AIDS. In contrast, SIV infection leads to the development of AIDS in rhesus macaques and these animals lack the activation of wound healing mechanisms to the same extent. Authors obtained rectal biopsies or necropsy tissue from AGM and RM animals during acute SIV infection and performed RNA-seq analysis to determine the gene expression profiles. Conserved gene signature analysis was applied to compare these data with previously published transcriptomic data sets across different species. Authors conclude that the activation of wound healing process in AGM preserves mucosal integrity, prevents inflammation mediated damage and halts disease progression. In summary, this manuscript reports a comprehensive comparative analysis of transcriptomic profiles of rectal tissues from AGM and RM during acute SIV infection and identifies gene signatures of protection from disease progression. The AGM gene data sets associated with the wound healing/repair and mucosal protection against virally induced damage provide a reference gene list and this will be of interest to HIV researchers.

There are several comments/questions that need to be addressed.

3.1 Viral load data are presented in plasma samples from AGM and RM animals. However, the gene expression data were obtained from rectal biopsies/necropsy tissue. What were the viral loads in rectal tissues? Were the levels of viral RNA similar or different in the rectal tissues from these animals?

REPLY: We have previously described the viral load in tissue in these RMs, by quantitative RT-PCR as well as mRNA-seq (Barrenas, 2014, J Virology). Here, we performed quantitative RT-PCR in AGMs, which showed no substantial difference in tissue viral load between the two host species.

3.2 AGM animals were infected with SIVsab and rhesus macaques were infected with SIVmac251. Authors should comment on the pathogenicity of these viruses and discuss the similarities/differences between these two viruses.

REPLY: We have now included details on the relative pathogenicity of these viruses in the Methods section.

Previously published papers (by the Authors and other investigators) have reported on gene expression profiles in mucosal tissues of AGM and RM animals and changes during SIV infection. In the current manuscript, Authors have compared the gene expression profiles across several species based on the published data sets. In this regard, Authors have expanded the depth of the gene expression analysis and have identified the wound healing process as an important mechanism. This is definitely a comprehensive rectal mucosal gene expression analysis of AGM and RM responses to acute SIV infection. By defining the previremic, viremic and set point stages, Authors have increased the number of animals per each group and this approach has strengthened the data analysis.

3.3 It is not clear whether rectal tissue samples were longitudinally from the same animal overtime. If they are not from the same animal, what is the magnitude of variability in the spread of SIV infection in the rectal tissue of these animals?

REPLY: The tissues were collected upon necropsy, after each animal was sacrificed. The number of animals sacrificed at each time point are shown in figure 1A. Baseline samples were taken by pinch biopsies at least two weeks before challenge. Thus, the study includes one baseline sample and one necropsy sample from each animal. These details are now included in the methods section.

3.4 What kind of histopathologic changes are seen between AGM and RM animals? Although the general trend in the gene expression changes may be similar, an animal-to-animal variation is evident in the Figure 1D.

REPLY: Although several tissues were examined upon necropsy, all were found to be unremarkable.

3.5 Conserved gene signature analysis was applied to compare the acute SIV infection gene expression data sets from AGM and RM to the available transcriptomic data sets across different species and inflammatory conditions and to the pathways in axolotl

wound repair. This is an interesting approach. However, there have been several papers published on the gene expression analysis in AGM and RM and in HIV infection on the recovery during ART and in RM models with vaccine protection. It would have been helpful to compare the acute SIV gene network and wound repair genes in rectal tissue of AGM with non-progressive asymptomatic HIV or SIV infections and from vaccine protected animals.

REPLY: Most SIV vaccine and ART studies are based on longitudinal blood samples. The blood samples collected from these animals will be described in a future publication. While it would be extremely valuable to examine rectal tissues in these animals, collecting rectal tissues is an invasive procedure and is avoided in most studies. To date, we are not aware of published SIV vaccine or ART studies that include rectal tissues transcriptomics.

3.6 Authors found a wound healing signature in the AGM model that included monocyte regulated mucosal tissue immune remodeling. AGM rectal tissue had increased expression of myeloid cell recruitment associated gene expression (Fig. 3E). Does histopathologic analysis substantiate this observation by increased number of macrophages in the rectal tissue of the AGM compared to RM?

REPLY: We have carried out extensive follow-up experiments pertaining to macrophages. In summary, we found no substantial differences in macrophages levels between the two host species. However, we did find a stronger co-staining of macrophages marker HAM56 with the wound healing protein fibronectin, in AGMs. Together with the monocyte transcriptomic profiling, these analyses support a different macrophages phenotype in AGMs, although the number of macrophages in rectal tissues is similar in RMs. These results can be found in the new figure 6 and supporting figure S11-S13.

3.7 In Figure 6A, Ham56+ cells are shown in AGM rectal tissue. What about the RM rectal tissue samples? Are there any differences between these samples?

REPLY: See the previous response.

3.8 In the Figure 5C, the FN immunostaining in RM rectal tissue is difficult to compare with the AGM tissue. Comparable levels of nuclear stain and better images may help validate the differences.

REPLY: We have re-organized the figure to allow larger panels for the immunostaining images

Reviewer #4 (Remarks to the Author):

The authors aim to examine how natural hosts of simian immunodeficiency virus (SIV) avoid AIDS despite lifelong infection by comparing a natural SIV host, African green monkey (AGM) to an AIDS susceptible species, rhesus macaque (RM) on the transcriptomic level. The authors developed a systems biology approach, referred to as conserved gene signature analysis (CGSA), based on correlating expression signatures in acute SIV infection across informative transcriptomic datasets from the public domain across diverse species. They find that during acute SIV infection, AGMs rapidly activate evolutionarily-conserved regenerative wound healing mechanisms (fibronectin) in mucosal tissue and that AGM monocytes exhibited an embryonic development and repair/regeneration signature featuring TGF- β and concomitant reduced expression of inflammatory genes compared to RMs

Since my background is in bioinformatics, I will only comment on the computational merits of this manuscript. While the findings seem interesting and biologically relevant, several points must be addressed on the analytical side:

4.1 It seems that the authors are comparing a rectal samples after necropsy to blood samples, which is problematic. Maybe I'm missing something, but from the results section, it is unclear how the different tissue types are handled.

REPLY: The rectal tissues collected at necropsy were compared to baseline pinch biopsies taken two weeks or more before SIV inoculation. Here, blood samples were only used to measure viral load. We have added this explanation in the revised methods.

4.2 From the description of the results, it is unclear how many samples per group and what method was used for differential expression. Which cutoffs were applied for significance?

REPLY: Differential expression was carried out using the edgeR library. Differential expression was defined as having an FDR-adjusted p-value < 0.05 and an absolute fold change stronger than 1.5. We have added a clarification to the methods.

4.3 Was multiple hypothesis testing applied? The authors carry out many comparisons and this needs to be taken into account.

REPLY: All p-values were adjusted using false discovery rate (FDR).

4.4 Instead of the pairwise comparisons of differential expression, I would suggest looking at a time series analysis (longitudinal comparison) using linear mixed models.

We did consider time series (TS) analysis. However, given our low number of time points time series TS methods are sometimes outperformed by pairwise differential expression.

E.g., see <https://doi.org/10.1093/bib/bbx115>. Given that the same animals were not analyzed longitudinally, we opted for the pairwise DE. Note that the aim of this analysis was to create a pool of DE genes for further analysis by CGSA – our experience from this study is that CGSA is highly stable to slight variations in the input genes.

4.5 The public datasets that were chosen to be included in CGSA are based on the original comparison and pathway analysis, which is rather biased. The logic behind this is circular and the results are expected.

REPLY: The reviewer raises a very important point. Indeed, it is expected that the CGSA, which is based on the GO enrichment will find similar functional associations. However, the rule of CGSA is to dig deeper into the biological processes by adding the time dimension we used in the wound healing analysis. It also allowed us to specifically match up- and down- regulation between acute SIV infection and the reference dataset (see Fig 2D), something that is not possible with GO enrichment alone. The intention of the functional enrichment tests throughout the paper, was in equal measure a quality control step and biological analysis. We have emphasized this point in the revised manuscript.

4.6 The authors should explain how they are integrating data from microarrays and RNA-Seq and also across different organisms.

REPLY: We have added text to the Methods, explaining how mouse, rat and axolotl data was combined with the human mapped RM and AGM data.

4.7 Cell type deconvolution approaches could be helpful in further examining the role of monocytes.

We have performed cell type deconvolution in ImmQuant and made an extensive addition to the online supplement. We first assessed the predicted number of monocytes, which was suppressed during the first days after inoculation in RMs, in line with figure 6D. In AGMs, the monocyte quantity stayed near baseline after SIV infection.

We also found a set of cell types that increased after inoculation in AGMs and stayed elevated, like the wound healing signature. These cell types included megakaryocytes, and their progenitor cell types: megakaryocytic colony forming unit and hematopoietic stem cells. This is also in agreement with the hypothesis that resulted from our axolotl analysis: that AGMs have a stronger stem cell function which enables non-inflammatory wound healing.

To summarize, this was a very valuable to our study and supported our initial conclusions remarkably well.

4.8 The limitations of the presented approaches should be listed in the discussion.

The main limitation in our approach is using bulk tissue for transcriptomic profiling. This approach lacks resolution regarding cell types, tissue localization and so on. We have added this reservation to the Discussion section of the revised manuscript.

REVIEWERS' COMMENTS:

Reviewer #1 (Remarks to the Author):

A fine effort at revision. Well done.

Reviewer #2 (Remarks to the Author):

The authors have addressed each of my comments/suggestions in a way that is mostly satisfactory. The revised manuscript has clarified several points that we initially confusing or described in an incomplete way. The additional data adds nicely to the manuscript.

Reviewer #4 (Remarks to the Author):

The authors adequately addressed the informatics/technical concerns that were brought up.